# Implantable optical fibers for immunotherapeutics delivery and tumor impedance measurement

Ai Lin Chin [1,4], Shan Jiang [2,4], Eungyo Jang[1], Liqian Niu[1], Liwu Li[3], Xiaoting Jia [2✉] & Rong Tong [1✉]

Immune checkpoint blockade antibodies have promising clinical applications but suffer from disadvantages such as severe toxicities and moderate patient–response rates. None of the current delivery strategies, including local administration aiming to avoid systemic toxicities, can sustainably supply drugs over the course of weeks; adjustment of drug dose, either to lower systemic toxicities or to augment therapeutic response, is not possible. Herein, we develop an implantable miniaturized device using electrode-embedded optical fibers with both local delivery and measurement capabilities over the course of a few weeks. The combination of local immune checkpoint blockade antibodies delivery via this device with photodynamic therapy elicits a sustained anti-tumor immunity in multiple tumor models. Our device uses tumor impedance measurement for timely presentation of treatment outcomes, and allows modifications to the delivered drugs and their concentrations, rendering this device potentially useful for on-demand delivery of potent immunotherapeutics without exacerbating toxicities.

[1] Department of Chemical Engineering, Virginia Polytechnic Institute and State University, Blacksburg, VA, USA. [2] Bradley Department of Electrical and Computer Engineering, Virginia Polytechnic Institute and State University, Blacksburg, VA, USA. [3] Department of Biological Sciences, Virginia Polytechnic Institute and State University, Blacksburg, VA, USA. [4]These authors contributed equally: Ai Lin Chin, Shan Jiang. ✉email: xjia@vt.edu; rtong@vt.edu

mmune checkpoint blockade (ICB) antibodies against cyto-
toxic T–lymphocyte–associated protein 4 (CTLA-4) or pro-
grammed cell death 1 (PD-1) have demonstrated that
reactivating anti-tumor immune responses can lead towards
tumor regression[1,2], and these ICB antibodies have been
approved by the United States Food and Drug Administration
(FDA) for the treatment of a broad range of tumors[3–7]. Simul-
taneous blockade of both CTLA-4 and PD-1 reverses T cell
dysfunction and generates more durable anti-tumor immunity
than either therapy alone[5,8–11]. However, the response rate of ICB
therapeutics tends to be low[3,5,12,13]. Additionally, immune-
related adverse events can occur (especially when combined
ICB antibodies are used) and can sometimes be life
threatening[3,5,14,15].

In light of the disadvantages of these and other systemic
immunotherapeutics, intra- or peritumoral treatments have been
evaluated as alternatives[16,17]. This strategy not only generates a
local anti-tumor immune response but also drives inhibition of
systemic and distal tumors via the induction of immune
memory[18], and this inhibition enables immunological targeting of
disseminated malignancies. Clinically, local immunotherapy has
been proposed primarily for the treatment of unresectable tumors
or for post-surgical adjuvant therapy to prevent local
recurrence[19].

However, intratumoral injection of immunotherapeutics does
not prevent them from entering the systemic circulation and
dispersing to distal organs. In mice, intratumorally administered
cytokines[20,21] were rapidly cleared from the local injection site
and were detected in peripheral organs within minutes of injec-
tion. This phenomenon limits the maximum tolerated dose of
these agents because of the undesired widespread exposure and
off-target inflammatory symptoms. To address this problem,
various investigators have linked immunotherapeutic agents,
including ICB antibodies and cytokines, to collagen-binding
peptide domains for systemic administration due to the sub-
stantial amounts of collagen present in solid tumors[22–24]. These
modified ICB antibodies retain the anti-tumor effects of their
predecessors but have fewer adverse effects. However, the mod-
ified antibodies can also bind to normal collagen-containing tis-
sues (e.g., connective tissues, artery walls), which may limit their
efficacy and dose.

Another local administration strategy involves implanting
degradable scaffolds or hydrogels loaded with immunother-
apeutics, such as cytokines or ICB antibodies, to control the
localization and activation of dendritic cells[25,26] or T cells[27–29].
This strategy shows superior efficacy to bolus injections, but a
mechanism for dosage adjustment, either to lower systemic
toxicities or to augment therapeutic responses, is largely lacking.
Additionally, some hydrogels and scaffolds require tumor resec-
tion prior to implantation, owing in part to their relatively large
volumes. Recent progress of on-demand drug delivery[30–35]
inspires us to explore a strategy for local drug delivery via a
system that would integrate a feedback loop with convenient
means for timely presentation of treatment results and allow us to
adjust drug loading to potentiate immunotherapy without com-
pounded toxicity. Specifically, we envision that localized delivery
could be realized with an implantable miniaturized device with
both delivery and measurement capabilities, preferably lasting for
weeks at a time, because durable anti-tumor immunity requires
prolonged drug retention[36].

In this work, we engineer an implantable miniature optical
fiber device (IMOD) that can be used for local delivery of ICB
antibodies and for monitoring of clinical outcomes by tumor
impedance measurement over the course of a few weeks. We
choose impedance measurement because unlike other methods
such as electron or fluorescence microscopy, this reliable and

convenient technique is non-destructive, non-invasive, and not
limited by penetration depth[37]. Importantly, we demonstrate that
tumor impedance is well-correlated with tumor size in several
mice studies and is responsive to the therapeutic treatments[38–41].
We also include the option of photodynamic therapy (PDT) due
to its capacity to enhance anti-tumor immunity and prolong
intratumoral drug retention[42]. We discover that in multiple
tumor models, the combination therapy involving PDT and
localized delivery of ICB antibodies via IMOD cures or delays
tumor growth and elicits durable anti-tumor immune responses.

## Results

**Design and fabrication of IMOD and its use for in vitro drug
release.** The IMOD comprises of a tubular polycarbonate optical
fiber (outer diameter, 1.23 mm; inner diameter, 0.57 mm; light
transmission rate, >89%) with a thin chemical-resistant layer of
polyvinylidene difluoride bearing two embedded copper micro-
electrodes (diameter, 125 μm) for impedance measurement
(Fig. 1a)[43,44]. To load a hydrophobic drug onto the outer surface
of the fiber, we applied multiple alternating layers of a drug-
containing poly(lactic-co-glycolic acid) solution in tetra-
hydrofuran and a highly concentrated poly(lactic acid) (PLA)
solution (50 mg/mL in tetrahydrofuran), the purpose of which
was to prolong drug release (Fig. 1b)[45]. Note that length of the
optical fiber and dosage of the loaded drug could be adjusted to
suit their application. Additionally, the inner channel of the
resulting coated fiber could be loaded with hydrophilic molecules
(e.g., ICB antibodies) via capillary action (Fig. 1b)[46].

Prior to in vivo implantation, the microelectrodes embedded in
the fiber were connected to an integrated-circuit chip that was
used for impedance measurement (Fig. 1c). In addition, tubing
was inserted into the chip to connect to the inner channel of the
fiber, which allowed for on-demand reloading of drugs (e.g., ICB
antibodies) into the optical fiber. Specifically, the tubing could be
connected to a syringe containing ICB antibodies solution (Fig. 1c,
d); after each injection of the solution, the end of the tubing was
sealed with epoxy glue, which could be removed with scissors for
refilling of the tubing.

To test the drug-loading and drug-releasing capabilities of the
fibers, we encapsulated rhodamine B, which mimics hydrophobic
drugs, into the surface coating of the optical fiber (five cycles,
Fig. 1e); and we filled the inner channel of the optical fiber with a
solution of fluorescein isothiocynate-bovine serum albumin
(FITC-BSA) in phosphate buffered saline (PBS). The compounds
on the outer surface of the fiber and in the inner channel were
visualized by fluorescence imaging (Fig. 1e). When the fiber was
immersed in phosphate buffered saline at 37 °C, 90% of the
rhodamine was released within 12 h (Supplementary Fig. 1a). To
avoid direct contact and prevent mechanical effect, transwell
assays were set up to test the ability of tumor cells to uptake
rhodamine B released from the coated fibers; we found that
rhodamine B was effectively absorbed and retained by 4T1
murine mammary carcinoma cells (Supplementary Fig. 1b).

The layer-by-layer coating approach was also used with
verteporfin, an FDA-approved photosensitizer for PDT; specifi-
cally, more than 40 μg of verteporfin per centimeter of fiber could
be loaded. An optical fiber coated with alternating layers of
verteporfin and PLA released 83% of verteporfin after 12 h. For
slower verteporfin release, we coated the fiber with three more
layers of PLA and found that the release rate was substantially
lower than that of the fiber without the additional PLA layers, as
determined by high-performance liquid chromatography (Sup-
plementary Fig. 1c). 4T1 breast cancer cell death was observed
20 h after cells were incubated with a verteporfin-loaded fiber in a
transwell and subjected to 20 s of near-infrared (NIR) light, while

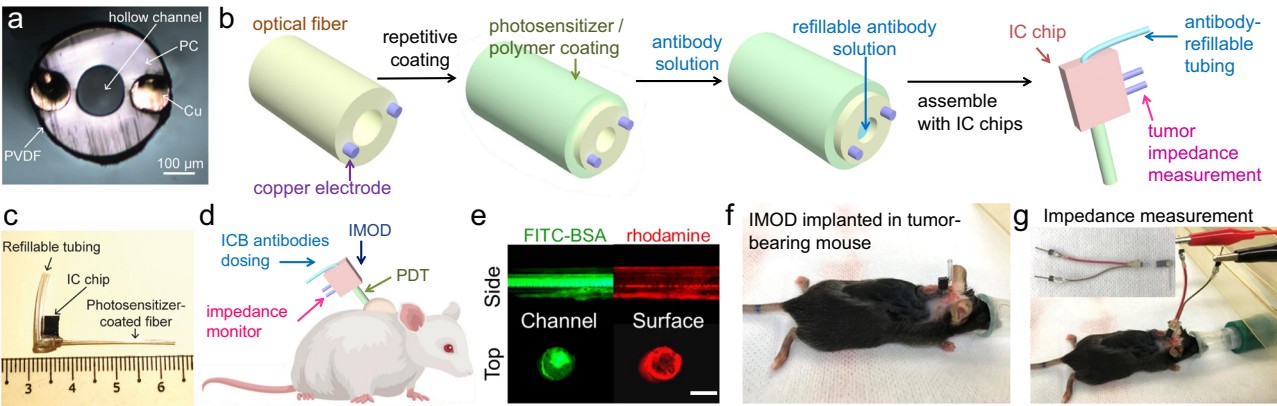

**Fig. 1 Design and fabrication of an implantable miniaturize optical fiber device (IMOD) for on-demand drug delivery and tumor impedance measurement. a** The cross-section image of an uncoated optical fiber with embedded electrodes for measuring tumor impedance. PC, polycarbonate; PVDF, polyvinylidene difluoride. **b** Procedure for coating the optical fiber surface with hydrophobic molecules (e.g., verteporfin or rhodamine) and filling the inner channel of the fiber with hydrophilic drugs (e.g., immune checkpoint blockade (ICB) antibodies or fluorescein isothiocyanate–bovine serum albumin (FITC-BSA)). The obtained fiber was assembled with an integrated-circuit (IC) chip to the IMOD device. **c** An image of IMOD with a refilling tubing and IC chip. **d** IMOD combines photodynamic therapy, immune checkpoint therapy, and impedance measurement, thus allowing for monitoring of treatment efficacy and adjustment of the antibody dosage to generate a sustained anti-tumor immune response while minimizing systemic toxicities. **e** Fluorescence imaging of an optical fiber coated with rhodamine B (red) and loaded with FITC-BSA (green). Scale bar: 500 μm. Data are representative of three repeated experiments. **f** Implantation of IMOD into a subcutaneous E0771 tumor in a C57BL/6 mouse. **g** Connection of IMOD in (**f**) via an electrical connector to a Gamry potentiostat for impedance measurement.

cells that were shielded from NIR light remained healthy, thus proving that verteporfin released from the fiber could be activated by NIR light (Supplementary Fig. 1d).

Ex vivo intratumoral diffusion experiments were performed by inserting rhodamine-coated fibers into murine 4T1 mammary tumors excised from BALB/c mice and incubating the fiber-bearing tumors in cell culture medium at 37 °C. The concentration of rhodamine in the excised tumors was retained for up to 10 days (Supplementary Fig. 2a, b). Fibers were also inserted into excised 4T1 tumors and filled with aqueous Cy5 dye conjugated to BSA (Cy5-BSA) to mimic intratumoral delivery of a hydrophilic agent. Fluorescence imaging showed that Cy5-BSA solution injected via the fibers quickly spread throughout the tumor mass and the Cy5 fluorescence signals persisted even after 48 h (Supplementary Fig. 2e). Taken together, these results showed that both hydrophobic small molecules and proteins could be sustainably delivered via the fiber.

**In vivo tumor impedance measurement.** With the fabricated IMOD, we first examined the feasibility of monitoring in vivo tumor growth by measuring tumor impedance. Since we suspected that tumor heterogeneity may lead to fluctuation of impedance values with tumor depth (Supplementary Fig. 3a), we carried out ex vivo impedance measurements of isolated 4T1 tumors with different sizes by implanting IMOD at numerous depths within the tumors. These experiments showed no significant differences between tumor impedance values at various depths, from the center to the periphery of tumors ranging in volumes from 100 to 1200 mm³ (Supplementary Fig. 3a).

To measure impedance in vivo, we implanted IMOD into a subcutaneous (s.c.) 4T1 tumor (50–100 mm³, ~5 mm diameter, Fig. 1f). Before implantation, the IMOD electrodes were tested in PBS to verify that they were functioning properly. During impedance measurement, the mouse was placed under anesthesia, and the implanted IMOD was connected to a Gamry electrochemical potentiostat (Fig. 1g). The entire measurement process lasted for approximately 10 min. Tumor impedance varied with frequency of current as shown in the Bode phase impedance plot in Supplementary Fig. 3b. At a low frequency (1 kHz), impedance

did not vary significantly with tumor size (Supplementary Fig. 3c). In contrast, at a frequency of 10 kHz, which is consistent with the frequency recommended in literature[47], normalized impedance increased linearly with increasing normalized tumor size ($R^2 = 0.81$, $n = 6$, Fig. 2a). Importantly, we were able to use IMOD to monitor tumor impedance signals over the course of 3–4 weeks of tumor growth. The ability of IMOD to record tumor impedance signals in other types of tumors was also evaluated, and similar linear relationships between impedance and tumor size were observed in s.c. E0771 breast tumors (C57BL/6 mice, $R^2 = 0.80$, $n = 8$, Fig. 2b), s.c. CT26 colon tumors (BALB/c mice, $R^2 = 0.83$, $n = 7$, Fig. 2c), and s.c. B16F10 melanoma tumors (C57BL/6 mice, $R^2 = 0.91$, $n = 5$, Fig. 2d).

**Effect of PDT on intratumoral drug retention.** The ability of the optical fiber in IMOD to transmit light enabled us to use PDT, which is an FDA-approved, minimally invasive therapeutic modality[48] in which an inactive light-sensitive molecule, called a photosensitizer, reacts with oxygen upon light irradiation to form reactive oxygen species that are cytotoxic to tumor cells[49]. Treatment of murine tumors by means of localized PDT was reported to increase anti-tumor immunity by releasing antigens and inflammatory cytokines from dying tumor cells[49,50]. Several studies using the combination of ICB and PDT, in which various photosensitizers or nanoparticles were administered, suggested that this combination shows enhanced anti-tumor efficacy relative to individual treatments[51–54]. Notably, enhanced tumor vessel permeation of therapeutics has been observed in PDT-treated mice and has been utilized to improve intratumoral accumulation of nanoparticles and drugs[55–57].

We then assessed whether PDT could improve intratumoral retention of ICB antibodies. We selected Cy7-BSA to mimic ICB antibodies because fluorescently labeled BSA is easy to prepare on a milligram scale for imaging studies. In addition, the molecular weight of BSA (55 kDa) is close to the molecular weights of anti-PD-1 (~40 kDa) and anti-CTLA-4 (~35 kDa). We administered Cy7-BSA via IMOD 4 h after PDT (i.e., verteporfin administration and light irradiation) and monitored the distribution of labeled Cy7-BSA by means of a whole-body imaging system. The

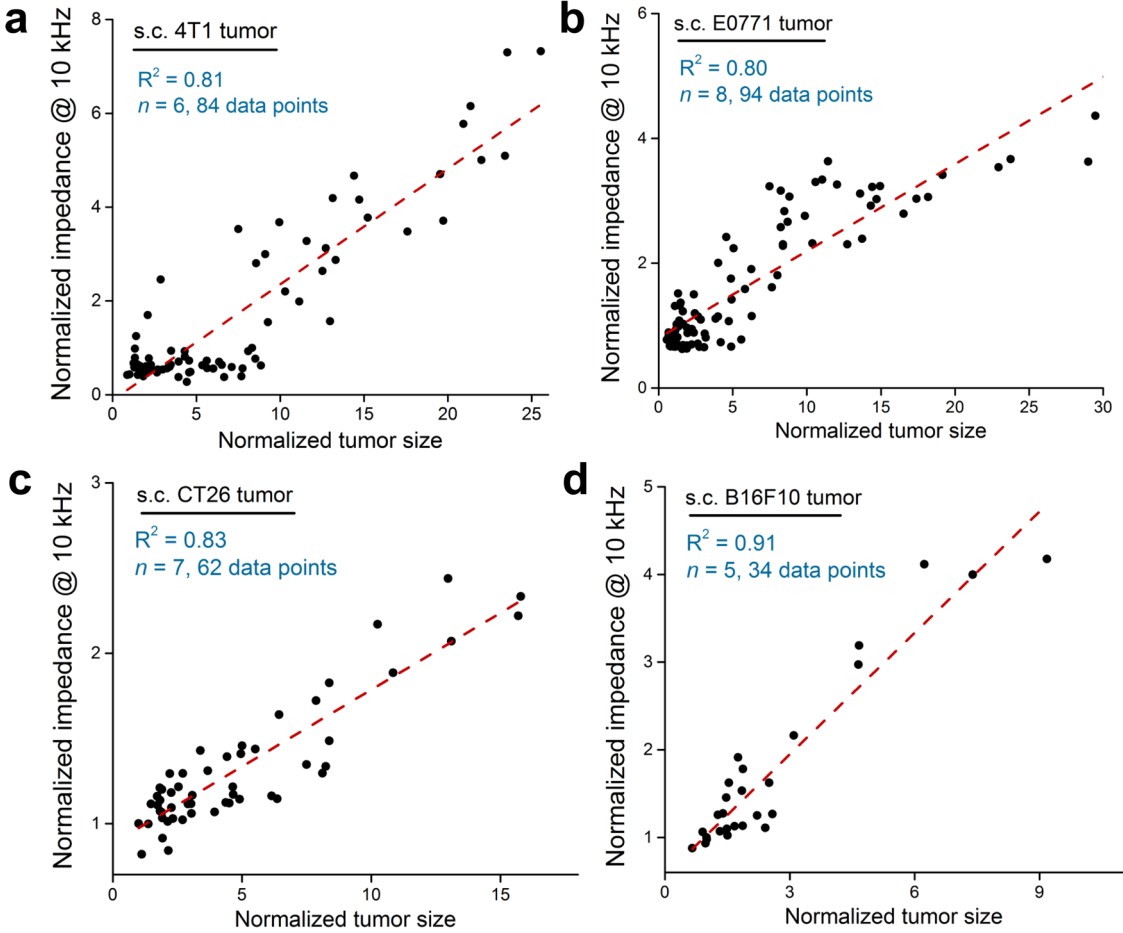

**Fig. 2 In vivo tumor impedance measurements using IMOD.** The linear regression relationships (red dashed lines) between normalized tumor size and normalized impedance at 10 kHz observed for subcutaneous **a** 4T1, **b** E0771, **c** CT26, and **d** B16F10 tumors. Normalized value is calculated based on the starting value of the measurement (set as 1 for both size and impedance readings at 10 kHz).

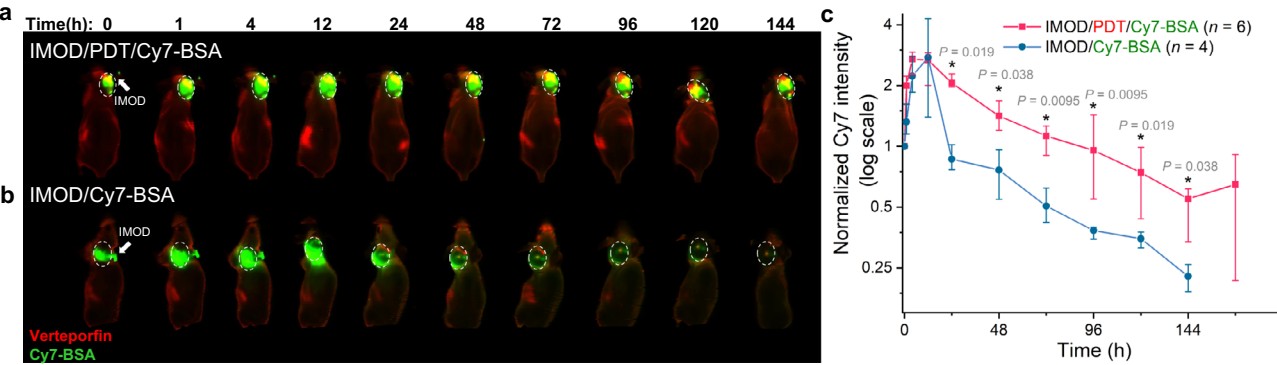

**Fig. 3 Effect of photodynamic therapy (PDT) on intratumoral retention of proteins.** Representative whole-body fluorescence images of **a** 4T1 tumor–bearing nude mice with an implanted IMOD (white arrow) that received PDT (red, −4 h) and Cy7–bovine serum albumin (BSA) via IMOD (0 h). **b** 4T1 tumor–bearing nude mice that received Cy7-BSA via IMOD (0 h). In both panels, the tumors are indicated with white dashed circles. **c** Time course of normalized intratumoral Cy7 intensity over 1 week (n = 4–6) for the mice shown in (**a**) and (**b**). Data are medians ± quartiles. Asterisks indicate P < 0.05, determined using Mann–Whitney U-tests.

fluorescence signal of Cy7-BSA was observable in the tumor for approximately 1 week (median half-life, 33.8 h; Fig. 3a, c). Conversely, Cy7-BSA administered through IMOD without prior PDT was quickly cleared from the tumors (median half-life, 8.9 h; Fig. 3b, c). Additionally, previous studies showed that PDT could cause a reduction in tumor blood flow rate[58–60], which may also attribute to the prolonged retention of Cy7-BSA. Note that prior

to PDT (4 h after implantation of IMOD), verteporfin stayed mainly around the tumor site (Supplementary Fig. 2f).

**PDT plus sustainable delivery of ICB antibodies via IMOD showed better in vivo efficacy than either treatment alone.** We evaluated the administration of PDT and ICB antibodies via IMOD in three s.c. syngeneic tumor models: E0771 breast tumors

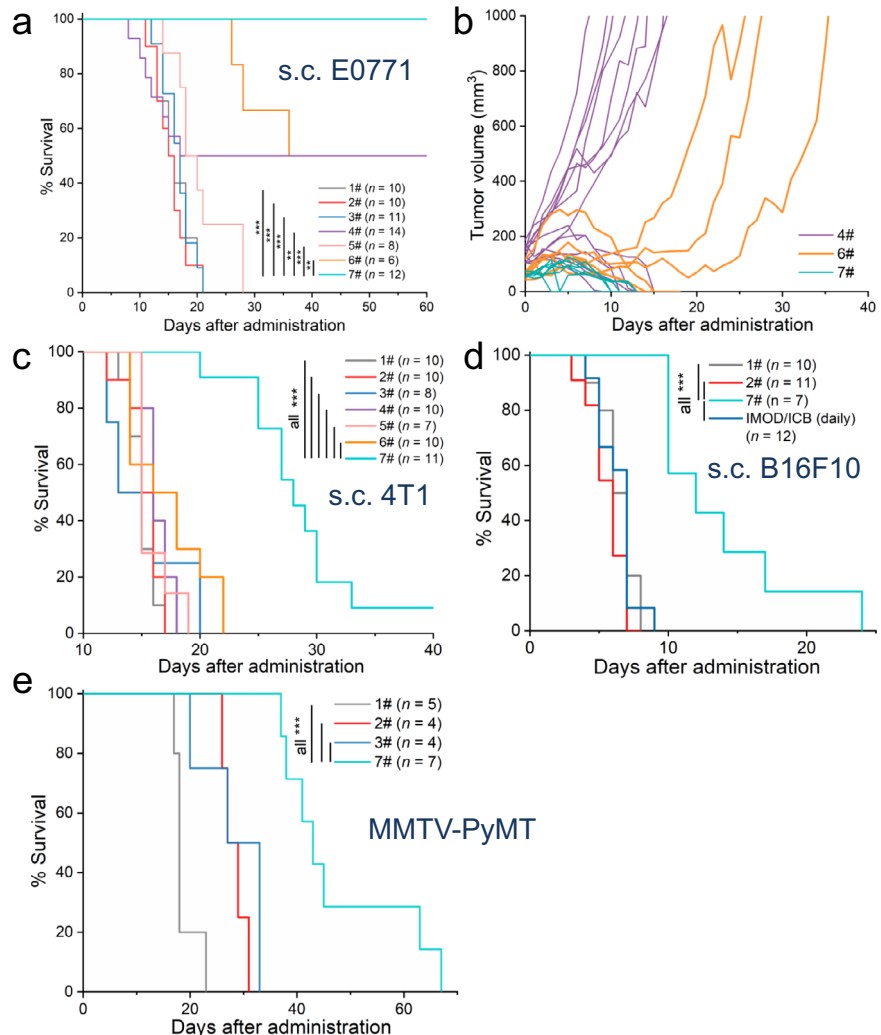

**Fig. 4 Effects of IMOD combining PDT and daily administration of ICB antibodies on tumor growth and survival in mice. a, b** Treatments performed in subcutaneous (s.c.) E0771 tumors in C57BL/6 mice ($n = 6$–14). Treatment regimens in (**a**), (**c**), (**d**), and (**e**): 1#, untreated; 2#, IMOD/PDT (q3d × 2); 3#, ICB (i.p. q3d × 4); 4#, ICB (i.t. q3d × 4); 5#, ICB (i.p. daily); 6#, IMOD/PDT (q3d × 2) + ICB (i.p. q3d × 4); 7#, IMOD/PDT × 2/ICB (daily). Abbreviation: i.p., intraperitoneal; i.t., intratumoral; q3d × 4, every three days for four times. IMOD/PDT × 2/ICB (daily): treatment with PDT (q3d × 2) and ICB antibodies (daily) via IMOD. For additional treatment groups, see Supplementary Fig. 4e. **c** Treatments performed in s.c. 4T1 tumors in BALB/c mice ($n = 7$–11). **d** Treatments performed in s.c. B16F10 tumors in C57BL/6 mice ($n = 7$–12). **e** Treatments performed in MMTV-PyMT transgenic female mice with spontaneous breast tumors ($n = 4$–7). In (**a**), (**c**), (**d**), and (**e**), statistical significance was determined using log-rank tests. * $P < 0.05$, ** $P < 0.01$, *** $P < 0.001$. Source data and $P$ values in a, c, d, e are provided in the Source data file.

in C57BL/6 mice, 4T1 breast tumors in BALB/c mice, and B16F10 melanoma tumors in C57BL/6 mice ($n = 6$–14, Fig. 4). Tumor cells ($10^5$) were injected subcutaneously into the back of the neck of mice and were allowed to grow to volumes of ~50 mm³ before initiation of treatment. The outside of the fiber of IMOD was coated with verteporfin (5 μg, which is equivalent to a dosage of 0.25 mg/kg verteporfin injected intravenously per mouse)[55], and IMOD was implanted into the tumor. Four hours after implantation, the tumor was irradiated via the fiber (62 mW/cm² for 20 s, see Methods) to activate the released verteporfin. ICB antibodies, anti-CTLA-4 and anti-PD-1 (5 mg/kg each)[61], were administered daily through the refillable tubing of IMOD. Light irradiation for PDT was applied again 3 days later, in order to activate the released verteporfin from IMOD (referred to as IMOD/PDT×2/ICB (daily)). This combination regimen induced robust tumor regression and cured all the E0771 tumor-bearing mice over the course of 60 days (Fig. 4a, b). In contrast, the intratumoral administration of ICB antibodies or PDT alone elicited weaker therapeutic responses than the combination

regimen ($P = 0.0051$ and $0.0000004$ by log-rank test, respectively, Fig. 4a). A partial tumor reduction response was observed when PDT (via IMOD) was combined with intraperitoneally administered ICB antibodies (q3d × 4, every 3 days for four times, $P = 0.0069$ by log-rank test compared to the combination regimen, Fig. 4a, b), indicating the importance of local administration of ICB antibodies. We note that it is impracticable for daily intratumoral administration to be performed on tumor-bearing mice; serum cytokines, including tumor necrosis factor–α and interferon-γ, and alanine aminotransferase (a liver damage marker) modestly increase upon repetitive intraperitoneal administration of ICB antibodies in mice[17,22]. The combination regimen of PDT and daily administered ICB antibodies via IMOD also displayed therapeutic effects in mice bearing 4T1 and B16F10 tumors, prolonging their survival relative to that of a group receiving only intratumorally administered ICB antibodies ($P < 0.001$ by log-rank test, Fig. 4c, d). Notably, despite the high response rates, the combination regimen had minimal systemic toxicity; neither the E0771 tumor-bearing mice nor the 4T1

tumor-bearing mice showed weight loss (Supplementary Fig. 4b, c). In addition, all mice that were cured of E0771 s.c. tumors by the combination regimen rejected a rechallenge with $10^5$ E0771 cells inoculated into the right flank on day 80. The mice that rejected the tumor rechallenge exhibited a significant increase in both effector and central memory CD8 + T cells in the lymph nodes on day 90 ($P = 0.001$ and 0.005, respectively; $n = 6$ or 7, Supplementary Fig. 5e, f), indicating that this combination therapy resulted in the formation of effective systemic immunological memory[62]. We then evaluated the anti-metastatic effect by intravenously injecting $10^6$ E0771 tumor cells into mice that were cured of E0771 s.c. tumors by the combination regimen on day 60 (Supplementary Fig. 6a). Remarkably, all mice that were previously cured by the combination regimen survived over 4 weeks after the aforementioned cell administration and no lung metastasis was found (Supplementary Fig. 6b, c). The substantial increase in both effector and central memory CD8+ T cells in the lymph nodes of these mice (Supplementary Fig. 6f, g) suggested that the combination regimen could induce long-term immunologic memory against tumors.

To extend our results beyond transplantable tumor models, we tested female FVB/N-Tg(MMTV-PyVT)634Mul/J mice (also known as MMTV-PyMT, mouse mammary tumor virus–polyomavirus middle T antigen) which spontaneously develop highly invasive ductal carcinomas in all 10 mammary fat pads with a high frequency of lung metastases[63]. When the first tumor on MMTV-PyMT mice had reached around 50–100 mm³ by 8 weeks of age, we started the treatment by implanting IMOD into the tumor site. The mice treated with the combination regimen had significantly lower total tumor burdens (Supplementary Fig. 7) and prolonged survival time compared to groups receiving either PDT or ICB antibodies ($P < 0.01$ by log-rank test, $n = 4$–7, Fig. 4e). These data indicated that our method could be applicable to genetically engineered mouse models of breast tumor that recapitulates virulent human breast cancer promoted by oncogenes[63].

**Tumor impedance measurement responds to the ICB antibody treatment via IMOD.** Next, we determined whether tumor impedance measured via IMOD would respond to the administered treatment—in other words, whether the tumor impedance readings would synchronously respond to the reduction in tumor size. First, we confirmed that tumor impedance measured by IMOD responded to treatment with intraperitoneally injected ICB antibodies (q3d × 3) in mice with s.c. CT26 tumors into which IMOD was implanted. We chose the s.c. CT26 tumor model because these mice were cured in 25 days by intraperitoneally injected ICB antibodies (q3d × 3, Supplementary Fig. 4a). Importantly, we found that the impedance readings obtained via IMOD correlated with the change in tumor size ($n = 5$, Fig. 5a, Supplementary Fig. 8a). Encouraged by these results, we then determined whether IMOD could be used for both impedance measurement and ICB antibody delivery in s.c. E0771 tumor model. The ICB antibodies were administered every 2 days and impedance values were recorded before injection of the antibodies. We found that in all of the treated mice, the decrease in E0771 tumor sizes elicited by the antibodies could be detected from the impedance measurements ($n = 6$, Fig. 5b, Supplementary Fig. 8b). These results demonstrated that we could achieve sustained delivery of ICB antibodies while simultaneously measuring impedance to monitor tumor growth.

**Combination therapy via IMOD impacts tumor immune infiltration.** Previous investigation of the mechanism by which ICB antibodies exert their effects has suggested that blockade of CTLA-4 and PD-1 checkpoints via systemic administration can reverse T cell dysfunction by increasing effector-to-suppressor cell ratios and producing proinflammatory cytokines such as interferon-γ (IFN-γ) and tumor necrosis factor-α (TNF-α)[8]. Encouraged by the improved anti-tumor efficacy we observed through localized combination therapy (PDT + ICB antibodies administered via IMOD), we examined the cellular infiltrates in treated E0771 tumors to improve our understanding of the tumor immune response elicited by the treatment. On days 9 and 10, leukocytes were extracted from E0771 tumors for flow cytometry analysis (for mice receiving combination regimen, the collection of tumor started on day 8 while tumor sizes were ~ 100 mm³). Mice treated with the combination regimen via IMOD showed higher numbers and percentages of CD8+CD3+ effector T (T_eff) cells in tumor infiltrating lymphocytes (TILs) ($P = 0.00009$, Fig. 6a, $n = 6$–14; Supplementary Fig. 10a; also see immunohistology images of intratumoral CD8+ cells in mice receiving the combination regimen in Supplementary Fig. 10e). Although we observed no significant difference in the fractions of CD4+CD3+ T_eff cells in TILs or the fractions of regulatory CD4+CD25+Foxp3+ T (T_reg) cells in total CD4+ TILs in most groups (Fig. 6b, c), the fractions of immunosuppressive myeloid-derived suppressor cells (MDSC, CD11b+ Ly6G⁻ Ly6C^hi) in TILs was significantly lower in mice that received the combination treatment ($P = 0.0008$, Fig. 6d). Importantly, we found a significantly higher fractions of cells with the T effector memory phenotype (T_EM, CD44^hi CD62L^lo) in intratumoral CD8+ T cells in mice that received the combination regimen (Fig. 6e), which is consistent with previous reports regarding administration of ICB antibodies[64,65] and ICB antibodies combined with radiation[66]. Notably, both the fractions of T_EM cells and cells with the central memory phenotype (T_CM, CD44^hiCD62L^hi) in CD8+ cells in tumor-draining lymph nodes increased in mice receiving combination regimen (Supplementary Fig. 11f, g). Furthermore, the ratio of CD8+ T_eff cells to T_reg cells, CD8+CD44+ cells to T_reg cells, and CD8+ T_eff cells to MDSC were substantially higher in mice that received the combination regimen than that in the other groups (Fig. 6f–h). These observations indicated that the higher efficacy of the combination therapy administered via IMOD was partially due to enhanced infiltration of T_eff cells and attenuation of suppressive immune cells. We also examined TILs in 4T1 tumors (Supplementary Figs. 12 and 13). Although no significant difference was observed in the fraction of CD8+ T_eff cells in TILs between the combination and untreated groups (Supplementary Fig. 13a), the fraction of T_EM cells in CD8+ cells was higher in the combination group than in other groups (Supplementary Fig. 13e). Moreover, the ratio of CD8+CD44+ cells to T_reg cells in the combination group was significantly higher than that in the other groups (Supplementary Fig. 13h).

Given that a significant proportion of CD8+ TILs were found to be PD-1+CTLA-4+ (Fig. 6i), a status that is associated with CD8 + T cell exhaustion[64,67,68], we examined the status of CD8+ TILs to find out whether our localized combination therapy could rescue the proliferation of tumor-reactive CD8+ cells. We sorted CD8+ TILs from E0771 tumors and performed ex vivo stimulation to the isolated CD8+ T cells to analyze the intracellular levels of cytokines in pre-activated T cells. The combination therapy administered via IMOD resulted in an enhanced proliferative level (ki-67+) of CD8+ T cells compared to that of other groups (Fig. 7a). Indeed, both proliferation (ki-67+) and cytotoxicity (IFN-γ, TNF-α, granzyme B, and IL-2) were significantly higher in the CD8+ TILs in the combination therapy group than the untreated group and the groups that received a single regimen (Fig. 7b–e). Notably, the substantial increase of proliferative polyfunctional ki-67+CD8+ TILs

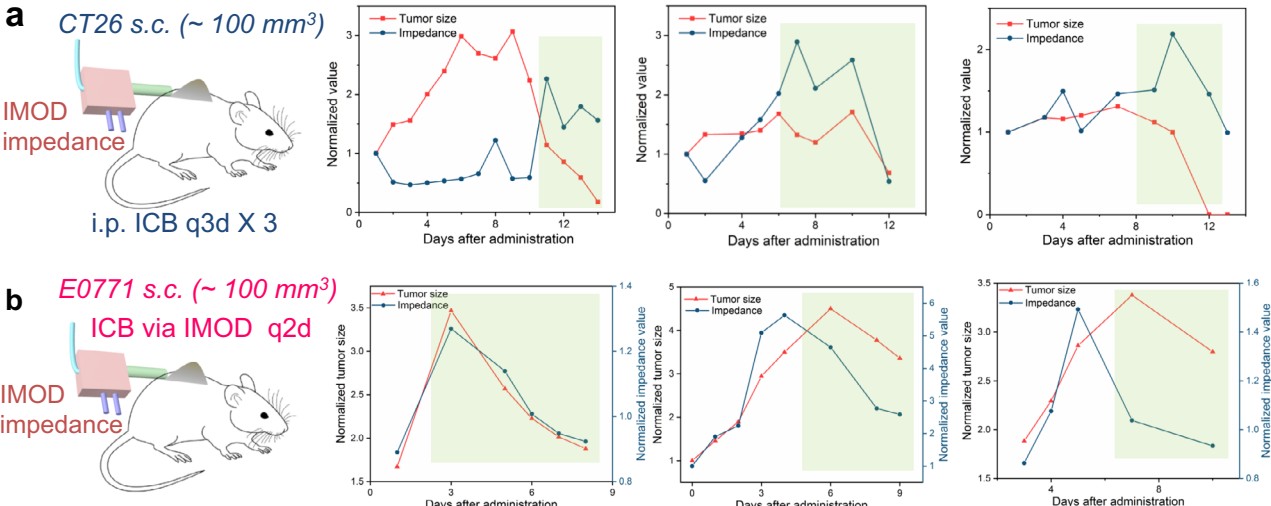

**Fig. 5 Correlation between tumor shrinkage and decreased impedance value (highlighted in light green). a** Representative results for treatment of s.c. CT26 tumor by intraperitoneally (i.p.) injections of ICB antibodies, with tumor impedance measured by an implanted IMOD. **b** Representative results for treatment of s.c. E0771 tumor by injection of ICB antibodies every 2 days via an implanted IMOD. Tumor impedance values were measured via IMOD before administration of treatment. Detailed results for both experiments are provided in Supplementary Fig. 8.

(IFN-γ+TNF-α+ and IFN-γ+IL-2+) elicited by the combination therapy was likely associated with improved relapse-free survival ($P < 0.001$, Fig. 7f, g), which is consistent with the results of many clinical and preclinical studies[69–72].

We then determined which subsets of CD8+ exhausted TILs changed in response to the combination therapy and evaluated their frequencies in TILs. We found significant increase of the fraction of progenitor exhausted CD8+ TILs (PD-1+CD44+Slamf6+Tim-3−), which is reported to persist over long-term with polyfunctionality, in the group receiving the combination therapy via IMOD compared to the groups receiving ICB antibodies ($P < 0.001$, Fig. 7h)[73]. Additionally, the fraction of terminally exhausted CD8+ TILs (PD-1+CD44+Slamf6−Tim-3+), which are often short-lived and differentiated from progenitor exhausted cells, was substantially lower in the group that received the combination therapy via IMOD than other regimens (Fig. 7i). Furthermore, we found that the dysfunctional subsets of PD-1+CD38+CD8+ and PD-1+LAG-3+CD8+ TILs[74–76] were significantly lower in the group that received the combination therapy via IMOD than in the groups that received ICB antibodies ($P < 0.001$, Fig. 7k-l). Thus, the combination therapy may have reversed the dysfunction of exhausted CD8+ TILs and maintained the polyfunctionality of progenitor exhausted CD8+ TILs to sustain a durable anti-tumor response.

## Discussion

In this study, we demonstrated that IMOD allowed us to simultaneously measure tumor impedance and deliver ICB antibodies. Recognizing that quickly and conveniently determining the efficacy of on-demand drug delivery has long been a challenge, our miniature device exhibited the potential to be useful for on-demand local drug delivery with the versatility to allow changes in the delivered drugs and their concentrations while possibly decreasing systemic toxicities (Supplementary Figs. 16 and 17). Combining PDT with delivery of ICB antibodies via IMOD resulted in a durable anti-tumor response, whereas administration of either treatment separately or in combination via different administration routes was often less effective. Importantly, we found that treatment-induced shrinking of tumor volume could be detected by monitoring tumor impedance via IMOD. To improve our understanding of the effects of tumor tissue growth

on tumor impedance, we proposed a modified circuit model to fit our experimental results (Supplementary Figs. 18–19, Supplementary Table 1). Data simulated with this model fitted well with our experimental impedance data and indicated that the increase in paracellular resistance and transcellular impedance due to increased tumor size may contribute to the proportional increase of measured impedance (see Circuit Model Discussion section in Supplementary Information). Notably, impedance measurements via IMOD were simple to carry out, and we envision that other sensors could be adapted to facilitate rapid in situ monitoring of physiological changes in tumors.

To date, therapies that combine PDT with ICB antibodies have been achieved by using light-responsive nanoparticles[51,52,54]. However, the detailed mechanism by which these therapies exert their effects has not been thoroughly studied. Prior work indicates that PDT may elicit an anti-tumor immune response by recruiting CD8+ cells or neutrophils[49,53,77]. We found that PDT had modest effects on CD8+ Teff cell responses in tumors (Fig. 6a and Supplementary Fig. 13a) but resulted in significantly higher levels of proliferative CD8+ T cells than treatment with ICB antibodies ($P = 0.004$ and $0.0004$, compared with groups receiving intraperitoneally or intratumorally injected ICB antibodies, respectively, Fig. 7a). Additionally, treatment with ICB antibodies increased the number of CD8+ TILs (Supplementary Fig. 10a), but resulted in low levels of proliferative markers (compared with the levels in other groups, Fig. 7a) and high fractions of terminally exhausted subsets of CD8+ TILs. Dysfunctional PD-1+CD38+ and PD-1+LAG-3+ CD8+ TILs (Fig. 7k, l) with decreased cytokine levels were observed to increase in treatment with ICB antibodies compared to the combination regimen (Fig. 7b–e). The results of the combination therapy administered via IMOD showed that the decreased ratio of terminally to progenitor exhausted CD8+ cells (Fig. 7j) could contribute to improved proliferation and polyfunctionality of CD8+ cells in generating durable anti-tumor immunity[73,78–80]. This finding agrees with the results of recent studies indicating that patients responding to ICB therapy show increased fractions of progenitor exhausted CD8+ TILs[81]. Notably, we also found that mice which received the combination therapy via IMOD had substantially higher numbers of CD8+ TEM cells than mice in the other treatment groups. TEM cells are long-lived effector cells destined to become memory cells with durable anti-tumor

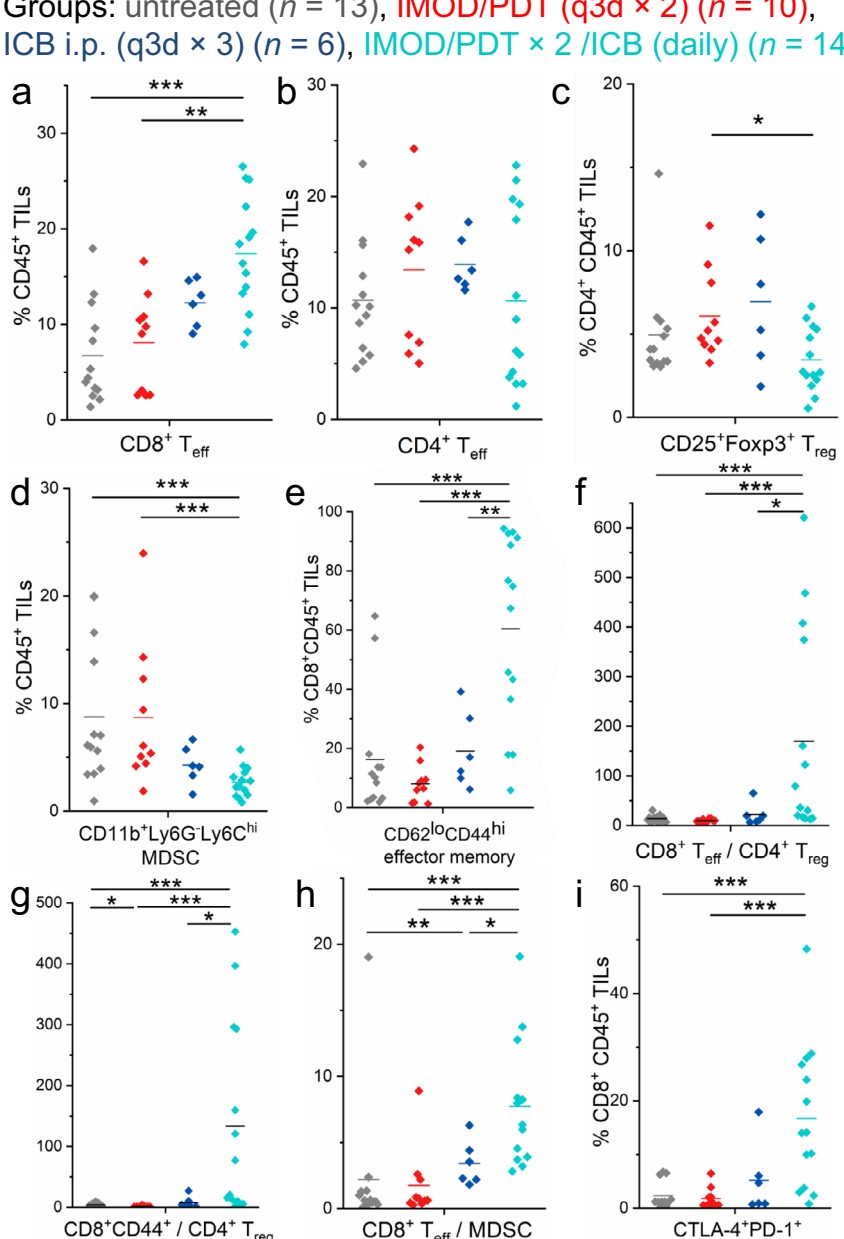

**Fig. 6 Effects of delivering a combination of PDT and ICB antibodies via IMOD on TILs in E0771 tumors.** The panels show data for E0771 tumors ($n = 6$–14) in C57BL/6 mice that were untreated or were treated with PDT (q3d × 2) via an implanted IMOD, or ICB antibodies (i.p. q3d × 3), or PDT (q3d × 2) and ICB antibodies (daily) via IMOD (referred to as IMOD/PDT × 2/ICB (daily)). Tumors were collected on day 9 or 10 after the start of treatment and were analyzed by flow cytometry (for mice receiving combination regimen, the collection of tumor started on day 8 while tumor sizes were ~100 mm$^3$). **a** Frequencies of CD8$^+$CD3$^+$ effector T (T$_{eff}$) cells in TILs. **b** Frequencies of CD4$^+$CD3$^+$ T$_{eff}$ cells in TILs. **c** Frequencies of CD25$^+$Foxp3$^+$ regulatory T (T$_{reg}$) cells in CD4$^+$ TILs. **d** Frequencies of CD11b$^+$Ly6G$^-$Ly6C$^{hi}$ myeloid-derived suppressor cells (MDSCs) in TILs. **e** Frequencies of CD44$^{hi}$CD62L$^{lo}$ effector memory cells in CD8$^+$ TILs. **f** Ratio of CD8$^+$ T$_{eff}$ cells to CD4$^+$CD25$^+$Foxp3$^+$ T$_{reg}$ cells. **g** Ratio of CD8$^+$CD44$^+$ T cells to CD4$^+$CD25$^+$Foxp3$^+$ T$_{reg}$ cells. **h** Ratio of CD8$^+$ T$_{eff}$ cells to MDSCs. **i** Frequencies of CTLA-4$^+$PD-1$^+$ cells in CD8$^+$ TILs. Statistical significance is determined by Mann–Whitney $U$-tests. * $P < 0.05$, ** $P < 0.01$, *** $P < 0.001$. Source data and $P$ values are provided in the Source data file.

immunity[81,82]. Clinically, increases in the frequency of intratumoral T$_{EM}$ cells have been observed in patients whose tumors respond positively to ICB antibodies[65]. Although the cause of the change in TILs in each treatment groups remains to be defined, local delivery of the combination therapy described herein promoted therapeutic response and anti-tumor immunity. Additionally, our studies also demonstrated the necessity of continuous administration of ICB antibodies in order to prevent CD8$^+$ TILs from terminal exhaustion (Fig. 7). The IMOD device is therefore

advantageous in practice to repetitively provide local administration of toxic agents to tumors since daily intratumoral injection using syringe is not feasible. More broadly, our current findings have substantial clinical relevance. By analogy to the demonstrated treatment efficacy in a genetically engineered mouse model with spontaneously arising tumors (Fig. 4e), the IMOD device could be applied in situ to vaccinate against human tumor driven by a strong oncogene before surgery in patients at high risk for the occurrence of metastatic disease or in patients genetically prone to

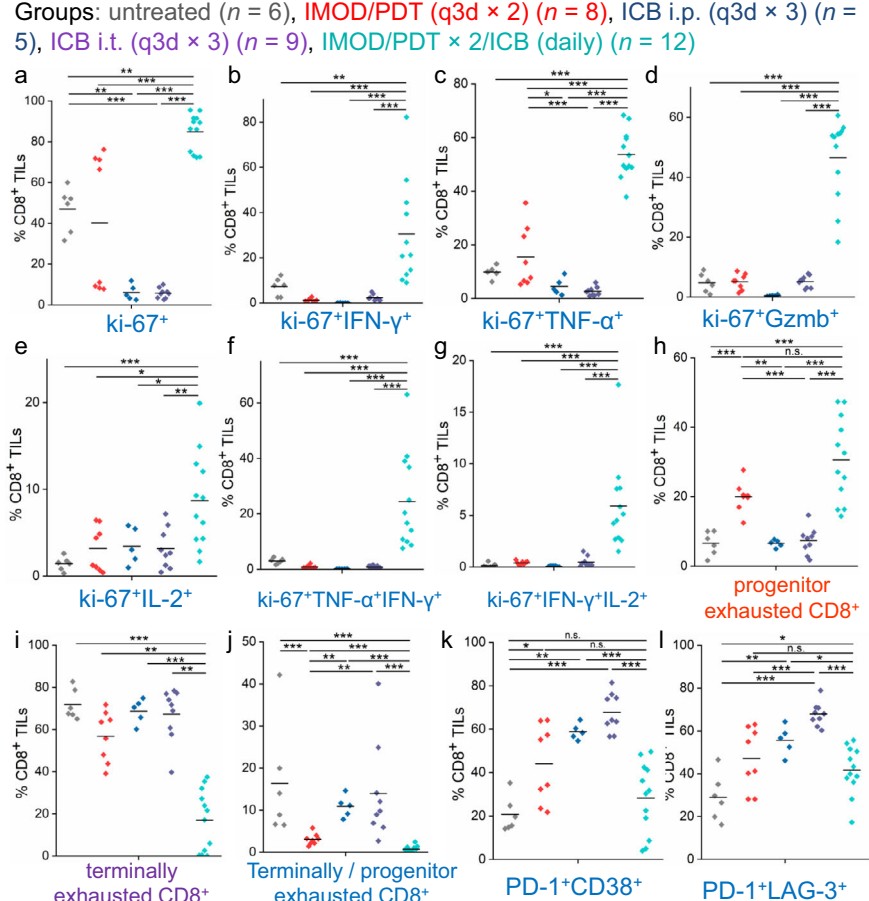

**Fig. 7 Analysis of CD8$^+$ TIL subsets that responded to combination therapy administered via IMOD in the treatment of E0771 tumors.** The panels show data for E0771 tumors ($n = 5$–12) in C57BL/6 mice receiving different treatments. **a** Frequencies of proliferative cells (ki-67$^+$) in isolated CD8$^+$ TILs. **b**–**g** Frequencies of cytokine production in isolated CD8$^+$ TILs after ex vivo stimulation. The graphs depict frequencies of proliferative (**b**) ki-67$^+$IFN-γ$^+$, (**c**) ki-67$^+$TNF-α$^+$, (**d**) ki-67$^+$Gzmb$^+$, (**e**) ki-67$^+$IL-2$^+$, and polyfunctional sets of (**f**) ki-67$^+$TNF-α$^+$IFN-γ$^+$, and (**g**) ki-67$^+$IFN-γ$^+$IL-2$^+$ cells in isolated CD8$^+$ TILs. **h**, **i** Frequencies of progenitor (**h**) and terminally (**i**) exhausted CD8$^+$ cells in isolated CD8$^+$ TILs. **j** The ratio between terminally and progenitor exhausted CD8$^+$ TILs. **k** Frequencies of PD-1$^+$CD38$^+$ cells in CD8$^+$ TILs. **l** Frequencies of PD-1$^+$LAG-3$^+$ cells in CD8$^+$ TILs. Statistical significance is determined by Mann–Whitney U-tests. *$P < 0.05$, **$P < 0.01$, ***$P < 0.001$. Source data and P values are provided in the Source data file.

developing second primary cancers, such as those with inherited mutations[83–85].

Two local immunotherapies have been approved by the FDA for cancer treatment: intravesicular injections of bacille Calmette–Guérin bacteria for the treatment of bladder cancer and intratumoral injections of an engineered virus (talimogene laherparepvec) for the treatment of melanoma. The number of preclinical studies of locally delivered immunotherapies has been increasing rapidly[24,27,28,86]. One concern regarding local administration of immunotherapies is the control of disseminated metastases. In this regard, we have demonstrated that durable anti-tumor immunity can be elicited against tumor rechallenge as a result of increased levels of CD8$^+$ T$_{CM}$ and T$_{EM}$ cells in lymph nodes (Supplementary Figs. 5, 6, and 11). Although the implantable nature of this system is not ideal, patients routinely undergo invasive and destructive procedures in an effort to cure cancers. In fact, patient acceptance of implantable systems (e.g. Implanon) is widespread, even in non-life-threatening situations[87]. Since this was a proof-of-concept study, we limited our investigation to ICB antibodies and we did not empirically optimize the dosing or timing of our treatments. We observed that although the combination therapy administered via IMOD delayed the growth of poorly immunogenic 4T1 and B16F10 tumors[88], the therapy was not curative (Fig. 4c, d, e). Demonstration of the applicability of

IMOD in deep-organ tumors that requires large animal models will be performed in the future while upcoming devices with wirelessly programmable signal measurement and communication functions are under development[34]. Our platform is amenable to the use of synergistic immunotherapeutics to potentiate anti-tumor effects in combination with ICB antibodies with the goal of treating "immune excluded" or "immune desert" tumors[71]. These systemically toxic agents including cytokines may benefit from localized delivery via IMOD, which has a feedback loop to monitor tumor response and deliver drugs on-demand.

## Methods

**Fabrication of electrode-embedded fibers and IMOD**. The fabrication of electrode-embedded fiber was based on reported procedures[44]. In brief, a layer of polycarbonate (PC) film with moderate thickness was rolled onto a teflon rod and consolidated in a vacuum oven to prepare a macroscopic preform that resembled the final fiber geometry. Two slots were machined 180 degree apart for electrode embedding, followed by a layer of polyvinylidene difluoride (PVDF) film till the proper diameter was reached. The preform was consolidated to enable a stable fiber drawing process. After the second consolidation, the teflon rod was removed from the preform and two copper wires were inserted into the two machined slots separately prior to the thermal drawing process. During the fiber thermal drawing process, copper wire was pulled from the coil which was mounted on a customized stage above the drawing furnace. As a result of a convergence fiber drawing, the outcoming fiber has a structure of one hollow channel in the center and two copper

wires opposite of each other outside the hollow channel (see Fig. 1a). To assemble the fiber with the integrated circuit (IC) chip for impedance measurement, copper wires in the fiber were manually exposed by a scalpel and electrically connected to external copper wires with conductive silver paint (SPI Supplies, West Chester, PA). These external copper wires were soldered to the headpins of the IC chip (Sullins Connector Solutions, San Marcos, CA) and epoxy was applied to protect the connection and avoid short circuits.

**Optical fiber coating and drug loading**. To prepare a polymer solution of hydrophobic molecules (e.g., rhodamine B) for fiber coating, a mixture of poloxamer (1000 mg, pluronic F127, Sigma-Aldrich), PLGA (330 mg, poly(D,L-lactide-co-glycolide), molecular weight: 30–60k, Sigma-Aldrich) in tetrahydrofuran (THF, 5 mL) was agitated vigorously in a 8 mL reaction vial until PLGA solution was homogenous and clear. Saturated rhodamine B/acetone solution was added into the PLGA/THF solution until solution was homogeneous and pink. Similarly, verteporfin (Cayman Chemical, Ann Arbor, MI)/PLGA solution was prepared by adding 2.5 mg/mL of verteporfin/THF solution and 5 mL PLGA solution into an 8 mL reaction vial.

To coat the fiber layer-by-layer with the polymer solution, a polycarbonate optical fiber was dipped into PLA/THF (100 mg/mL) solution and removed slowly so as to create a smooth coating. Next, it was placed into an air-tight chamber connected to vacuum. The coated fiber was dried under vacuum for 30 min before it was dipped into the rhodamine B / PLGA solution. The fiber was dried under vacuum again for 60 min. This was repeated until there were 11 layers of alternating polymer coatings (additional 1 PLA layer outside of rhodamine B/ PLGA). Verteporfin-coated fiber was prepared similarly. To extend the release of verteporfin, three additional layers of PLA coatings were coated onto the verteporfin fibers.

**Animal studies**. All animal work was conducted in accordance to the National Institutes of Health Guide for the Care and Use of Laboratory Animals under protocols approved by Virginia Tech Institutional Animal Care and Use Committee and Institutional Biosafety Committee (animal protocol numbers: 15-220, 18-120, 19-002, 21-029).

**Mice and cell lines**. BALB/c and C57BL/6 female mice (age 4–6 weeks from Jackson Laboratories or Envigo), nu/nu female mice (age 4 to 6 weeks from Charles River), FVB/N-Tg(MMTV-PyVT)634Mul/J male (age 4–6 weeks from Jackson Laboratories; JAX stock #002374) and FVB/NJ female mice (age 4–6 weeks from Jackson Laboratories; JAX stock #001800) were used and maintained according to approved animal protocol. MMTV-PyMT female mice with spontaneously developed breast cancer (age 6–7 weeks) were crosses of FVB/N-Tg(MMTV-PyVT) 634Mul/J male and FVB/NJ female mice, and genotyped before 3 weeks of age (Transnetyx). All mice were group-housed (5 mice per cage) and maintained under a regular light-dark cycle altered every 12 h with free access to water and food under pathogen-free conditions in a barrier facility at Virginia Tech. Murine breast cancer E0771 cell line was purchased from CH3 Biosystem (Amherst, NY, catalog # 940001); murine breast cancer 4T1 (catalog # CRL-2539), murine colon cancer CT26 (catalog # CRL-2638), and murine melanoma B16F10 (catalog # CRL-6475) cell lines were originally purchased from American Type Culture Collection (ATCC, Manassas, VA). All cells were maintained at 37 °C with 5% $CO_2$ in culture medium according to instructions, supplemented with 10% heat-inactivated FBS and penicillin/streptomycin (all from Life Technology, Grand Island, NY). All cells were tested to be free of mycoplasma.

**Subcutaneous tumor inoculation**. For subcutaneous (s.c.) inoculation of single B16F10, 4T1, E0771, and CT26 tumors, $5 \times 10^5$ cells in 50 μL of sterile PBS (1×) were subcutaneously injected into the back of neck of C57BL/6 or BALB/c female mice after hair was removed. Treatments were started when tumors reached ~50 mm³. The body weight and tumor size were measured every 1–2 days after the treatment started. Tumor length and width were measured with a digital caliper, and the tumor volume was calculated using the following equation: tumor volume = length × width × width/2[89]. Mice were euthanized when their tumor volumes reached a predetermined end point (1000 mm³) or when their body weights dropped over 10%. For tumor rechallenge studies, mice that overcame tumors were inoculated with the same amount of tumor cells ($5 \times 10^5$ cells) in the right flank and monitored for another 2 weeks. For evaluation of anti-metastatic effect, mice that overcame tumors were intravenously injected with $10^6$ E0771 tumor cells via tail-vein injection and monitored for another 4 weeks.

**IMOD implantation and treatments**. Mice receiving treatments through IMOD were placed under anesthesia, with s.c. injection of carprofen (5 mg/kg) as analgesia prior to implantation. Implantation of IMOD was started by making a small slit on the edge of tumor using sterile surgical scissors. The optical fiber end of IMOD was gently inserted into the tumor and the device was positioned under skin with refillable tubing remaining above skin level (see Fig. 1). Skin was pulled taut and adhered around device using 3 M Vetbond Tissue Adhesive. An ethanol solution (70%) was applied appropriately to maintain cleanliness of surgical wound. Intratumoral treatments of ICB antibodies via IMOD were administered in 100 μL

sterile PBS through refillable channel on IMOD. In total, 100 μg of anti-CTLA-4 (clone 9H10, BioXCell, Lebanon, NH; catalog # BE0146, diluted to 1 mg/mL) and anti-PD-1 (clone RMP1-14, BioXCell, catalog # BE0146, diluted to 1 mg/mL) were administered each per dose. Note that once the tumor had shrunk from large tumor size of over 100 mm³ to small size of below 75–100 mm³, the device was carefully removed from mice and the treatment was stopped.

For photodynamic therapy, verteporfin loaded fiber was trimmed to 1 cm per dosage and implanted into tumor. At 4 h post-implantation, the mouse was anesthetized with isoflurane. The tumor area was irradiated with near infrared light (Kessil H150-red LED light source of 34 W, 20 s, 600–700 nm) while the rest of the mouse body was covered with aluminum foil. For groups treated with IMOD/ PDT× 2, 3 days after the first treatment, mice were anesthetized with isoflurane and the tumor site was irradiated with near infrared light for 20 s in order to activate the released verteporfin from the device. For the group receiving i.p. or i.t. injected verteporfin (5 μg, equivalent to 0.25 mg/kg), light irradiation was also applied onto tumor site for 20 s at 4 h post-administration.

To measure the light intensity of IMOD, a two-centimeter long IMOD was coupled to a silica fiber patchcord (Thorlabs) which was air-coupled to the NIR light source (Kessil H150-red LED light source of 34 W, 600-700 nm). The light output was measured by a power meter (Thorlabs) with a photodetector (Thorlabs) attached, and the average power recorded was 62.4 mW/cm² leading to a light dose of 1.25 J/cm² that was delivered to the tissue in 20 s. To measure the light intensity for externally applied photodynamic therapy, the same light source (positioned away from the animal to avoid the heat's impact) and measuring power meter were used. For a typical exposed tissue area for light irradiation with diameter ~ 0.5 cm, the average power was 63.82 mW/cm², similar to that of IMOD.

**In vivo tumor impedance measurement**. For tumor impedance measurements, tumors were inoculated and allowed to form tumor mass of approximately 50-100 mm³, as described above. IMOD was implanted and impedance reading was recorded by connecting Gamry Interface 1000™ Potentiostat to IC chip on IMOD. Mice were placed under anesthesia while all impedance readings were taken. Treatments were given after impedance readings were measured for better consistency. For both measuring and dosing of drugs by IMOD, Pt / Cu were used to replace Cu / Cu electrodes due to its stability for impedance measuring over weeks. Normalized value was calculated based on the starting value of the measurement (set as 1 for both size and impedance readings at 10 kHz).

**Whole-body fluorescence imaging of mice**. Whole body fluorescence imaging of mice was performed with LI-COR Biosciences Odyssey Infrared Imaging System with emission wavelength at 700 nm (for verteporfin) and 800 nm (for Cy7-BSA). The nu/nu mice were first subcutaneously inoculated with $5 \times 10^4$ 4T1 cells in 50 μL PBS (1×) into the back of neck. Similar to the descriptions above, IMOD with verteporfin loaded optical fiber was implanted once tumor length was determined to be approximately 8 mm. The nu/nu mice were fed with alfalfa-free food for at least one week prior to the study in order to minimize the gastrointestinal background autofluorescence[90]. For imaging studies, 100 μg/mL Cy7-BSA in 30 μL sterile PBS (1×) was injected through the refillable tubing on IMOD into the 4T1 tumor after PDT irradiation. Fluorescence images of nu/nu mice were taken at designated time points. The analysis of the results was carried out using Origin software (Northampton, MA) by fitting the normalized fluorescence intensity (I) – time (t) curve into an exponential decay model according to Eq. (1).

$$I = a \cdot exp(-t/T_1) + b \qquad (1)$$

where a, b, $T_1$ are all constants fitted by Origin software. The starting point for the curve fitting was the peak value of the normalized fluorescence intensity, usually at 4 or 12 h. The obtained $T_1$ value was used to calculate the decay half-life τ based on Eq. (2).

$$\tau = ln(2) \cdot T_1 \qquad (2)$$

Note that the fluorescence intensity at t = 0 was normalized as 1.

**Flow cytometry analysis of tumor lymphocytes**. Tumors, lymph nodes and spleens were resected from mice, weighted and gently ground to generate single cell suspensions through a 70-mm cell strainer. Red blood cells were lysed with RBC Lysis Buffer (Biolegend, San Diego, CA, catalog # 420302). Cells were counted and resuspended in Cell Staining Buffer (Biolegend) and used for flow cytometry staining. Non-specific immunofluorescent staining was prevented by incubating cells with TruStain fcX™ (anti-mouse CD16/32) (clone 93, catalog # 101302, dilution 1:50, Biolegend) antibody in 100 μl volume for ~10 min on ice. Cell-surface staining with antibody was performed according to manufacturer's instructions (Biolegend) with a dilution ratio of 1/100 (also see reporting summary of this paper). Fluorescent antibodies (all from Biolegend) used included CD45 (clone 30-F11, catalog # 103108, dilution 1:100), CD3 (clone 17A2, catalog # 100220, dilution 1:100), CTLA-4 (CD152, clone UC10-4B9, catalog # 106312, dilution 1:100), CD11c (clone N418, catalog # 117348, dilution 1:100), CD49b (clone HMα2, catalog # 103518, dilution 1:100), PD-1 (CD279, clone RMP1-30, catalog # 109116, dilution 1:100), F4/80 (clone BM8, catalog # 123131, dilution 1:100), CD19 (clone 6D5, catalog # 115510, dilution 1:100), CD4 (clone GK1.5, catalog # 100460, dilution 1:100), Ly-6G (clone

1A8, catalog # 127608, dilution 1:100), CD11b (clone M1/70, catalog # 101262, dilution 1:100), CD25 (clone PC61, catalog # 102012, dilution 1:100), CD8a (clone 53-6.7, catalog # 100722, dilution 1:100), CD44 (clone IM7, catalog # 103008, dilution 1:100), CD62L (clone MEL-14, catalog # 104412, dilution 1:100), Ly-6C (clone HK1.4, catalog # 128016, dilution 1:100), CD38 (clone 90, catalog # 102722, dilution 1:100), Tim-3 (CD366, clone RMT3-23, catalog # 119704, dilution 1:100), LAG-3 (CD223, clone C9B7W, catalog # 125221, dilution 1:100), Slamf6 (Ly-108, clone 330-AJ, catalog # 134610, dilution 1:100), IFN-γ (clone XMG1.2, catalog # 505806, dilution 1:100), TNF-α (clone MP6-XT22, catalog # 506308, dilution 1:100), IL-2 (clone JES6-5H4, catalog # 503808, dilution 1:100), Granzyme B (clone QA16A02, catalog # 372214, dilution 1:100), ki-67 (clone 16A8, catalog # 652411, dilution 1:100), and FoxP3 (clone MF-14, catalog # 126404, dilution 1:100). Fixable live/dead cell discrimination was performed using Zombie Aqua™ Fixable Viability Kit according to manufacturer's protocol (Biolegend, catalog # 423102). For intracellular staining, cells were fixed and permeabilized with the True-Nuclear™ Transcription Factor Staining kit (Biolegend, catalog # 424401) following manufacturer's instruction before being stained with antibodies. All flow cytometric data collection was performed using BD FACSARIA™ flow cytometer (BD Biosciences, San Jose, CA) and analyzed using FlowJo software (Ashland, OR). Gating strategies are provided in Supplementary Figs. 9 and 15.

**CD8+ T cell isolation, activation and cytokine analysis**. Tumor CD8+ TILs were isolated using MojoSort™ mouse CD8 cell isolation kit (Biolegend, catalog # 480035) following manufacturer's instruction. Isolated CD8+ TILs were numerated and plated at a density of approximately $1 \times 10^6$ cells per well in a 96-well plate. The suspension was incubated with cell activation cocktail (containing brefeldin A, phorbol 12-myristate 13-acetate, and ionomycin; Biolegend, catalog # 423304) for 6 h at 37 °C with 5% $CO_2$ following manufacturer's instruction, followed by surface and intracellular flow cytometry staining to detect IFN-γ, TNF-α, other cytokines and cell markers.

**Immunohistological staining**. For immunohistological staining of tumor tissues, 5 mm formalin fixed, paraffin embedded tumor tissue sections were deparaffinized with xylene, rehydrated, and heated in sodium citrate H.I.E.R. (Heat Induced Epitope Retrieval, pH=6, Biolegend, catalog # 420902) for 30 min at 95 °C in a slide glass rack to perform antigen retrieval. Slides were blocked in 3% BSA for 30 min at room temperature, and stained with primary antibody rabbit anti-CD8 antibody (Abcam; catalog # ab203035, dilution 1:200, room temperature, 60 min), then Ultra Streptavidin HRP Kit (Biolegend catalog #929501) containing diaminobenzidine (DAB) chromogen was applied following the manufacture's protocol for visualization, in which brown staining represented presence of the targeted molecule. They were then counterstained with hematoxylin (Vector Laboratories, catalog # H-3401) for 5 min and rinsed with running tap water.

**Statistical analysis**. Statistical analysis was performed using Origin Software (Northampton, MA). Most comparisons between groups were assessed using Mann–Whitney $U$-test (two-sided). Kaplan–Meier survival curves were compared using log-rank test. Significance was represented as follows: *$P < 0.05$, **$P < 0.01$, ***$P < 0.001$, and not significant (n.s.). The $n$ values and specific statistical methods are indicated in figure legends.

## Data availability
The authors declare that all data supporting the current findings of this study are available in the main manuscript or in the Supplementary information. The source data are provided as a Source Data file with this paper. Other data are available from the corresponding author upon reasonable request.

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

## Acknowledgements

This work was supported by start-up funding from Virginia Polytechnic Institute and State University. A.L.C., E.J., and R.T. acknowledge support from American Chemical Society Petroleum Research Fund (57926-DNI-7) and National Science Foundation (CHE-1807911). S.J. and X.J. are grateful for the support of National Science Foundation (ECCS-1847436). We thank Melissa Makris (College of Veterinary Medicine, Virginia Tech) for her help on flow cytometry.

## Author contributions

A.L.C., S.J., X.J., and R.T. conceived the idea and designed the experiments. A.L.C., S.J., E.J., and L.N. performed the experiments. A.L.C., S.J., L.L., X.J., and R.T. analyzed the data and wrote the manuscript.

## Competing interests

A provisional patent (U.S. Patent Application No. 62/750,870) has been filed pertaining to the results presented in this paper by the authors (A.L.C., S.J., X.J., and R.T.). The authors declare no other competing interests.
