## [Peer Review File · Nature Communications]

Reviewers' Comments:

Reviewer #1:

Remarks to the Author:

This manuscript developed the implantable optical fiber to release photosensitizer/ICBs for combined therapy. The authors show that this implantable optical fiber could reflect the therapeutic efficacy via tumor impedance. In this work, the combination between PDT and ICBs seems to exert good therapeutic efficacy as well as immune response. However, this efficacy may be just because of the very high frequency of ICBs injection, and some major issues need to be addressed.

1. In figure 2, the authors didn't show how to calculate the normalized values, even in the experiment section. More importantly, the relationship between impedances and tumor sizes, but not their normalized values, should be more direct, convincing and meaningful. In addition, more discussions/explanations are necessary to illustrate the mechanism about the relationship between impedances and tumor sizes. Although the authors provided several references, it seems that the literature doesn't seem to support it.
2. In figure 4, daily treatment of IMOD/PDT/ICB (ICBs: 5 mg/kg) is in a really high frequency, especially considering the claim from the authors that PDT prolonged the retention of ICB even up to 144 h in tumor (figure 3, line 81). The combination of free photosensitizer and ICB is a required control to confirm the advantage of IMOD/PDT/ICB.
3. In figure 6, could the authors explain why anti-CTLA-4/anti-PD-1 in IMOD/PDT/ICB group could not block the CTLA-4/PD-1 in T cells, but increase the frequencies of CTLA-4+PD-1+ cells in CD8+ TILs?
4. Although the authors show the H&E images on day 7, a relative long term of biosafety of this device should be provided. Did the device induce the inflammation or cytokines in the interface between device and tissue?
5. In figure 5, tumor sizes would be better than their normalized values. Besides, did tumor impedance measurement respond to the ICB in terms of the mean value for five mice, but not just the value for individual mouse?
6. In figure 3, the fluorescence intensity in panel a at 0 h seems stronger than that in panel b. Can the authors show the absolute intensity, but not the normalized intensity in panel c?
7. In figure 1e, using rhodamine 6G to mimic hydrophobic drugs was not appropriate. The concentration of rhodamine 6G in water could be 1 mg/mL.
8. Please carefully check the manuscript, as there are many errors. Some descriptions in text are not consistent with the figures.

Reviewer #2:

Remarks to the Author:

The submitted manuscript describes IMOD (implantable miniature optical fiber device) as a technology to be used in the treatment solid tumors, via local delivery of both photosensitizer for photodynamic therapy (PDT) and antibodies for immune checkpoint blockade, as well as a tool to track tumor response (size) by impedance measurements. Data are presented on the immunological advantage of IMOD in stimulating anti-tumor immunity compared to either treatment as a standalone approach, and cured animals are demonstrated to reject a subsequent tumor rechallenge. A general comment relates to the explanation that is provided for why the combined approach is better than either modality as single treatment. It is stated that PDT acts to cause tumor vessel permeation (section on Effect of PDT on intratumoral drug retention), which can increase tumor concentrations of drugs. This is true, but the present study utilized intratumoral administration of both the photosensitizer and ICB antibody. These drugs were not delivered systemically (e.g., i.v.), making the role of vascular effects less certain. Further clarification of how vascular permeability would be altered by IMOD or an alternate explanation for the expected interaction between ICB and PDT in the context of IMOD would be valuable here.

Other specific comments include the following:

In supplemental Figure 2b, how are the images to be aligned with the photographs of Figure 2a? It

is not clear what Figure 2b is intended to add in addition to that shown in 2a.

Supplemental Figures 1c, 1d and 14 do not appear to be called out in the text. Certain panels of Supplemental Figure 5 are also not called out and the relevance of the b cell studies is not mentioned in the text.

What do the colors represent on Supplemental Figure 3a and 3b?

In Figure 2, it is stated that 6 mice were used for the measurements of panel a, 8 for panel b, 7 for panel c and 5 for panel d. This leads to confusion in the interpretation of the plots because tumor size does not seem to repeat itself as expected. Using panel a as an example, 84 data points from 6 mice suggests that there should be 14 data points plotted at each for 6 distant tumor volumes.

What is the injected dose of verteporfin that 5ug is equivalent to?

In Figure 5/Supplemental Figure 6, please further clarify what the green highlighting is used to indicate. Why do the first set of measurements often go in opposite directions within this green box?

Figure 4b show that E0771 tumors had resolved by 10 days after the initiation of treatment. How was this dealt with in subsequent immunophenotyping when tumors were collected at this same timepoint for flow cytometry? What was available to assay if tumors had resolved?

Under the section on Combination therapy via IMOD impacts tumor immune infiltration, the text refers to Figure 6h as the ratio of MDSC to Teff cells instead of the ratio of Teff to MDSC, as shown in the figure.

Supplementary Figures 15 and 16 demonstrate that IMOD was not associated with toxicities, but do not show that it reduced toxicities as claimed in the first paragraph of the Discussion. Comparison to a condition that caused toxicity was not performed, and the studies extended to only 7 days which is not a long period to justify that no toxicities were detected.

In reference to the following statement in the Discussion: "However, treatment with ICB antibodies increased the number of CD8+ TILs (Supplementary Fig. 8a) but also resulted in high fractions of terminally exhausted subsets of CD8+ TILs (Fig. 7i) and dysfunctional PD-1+CD38+ and PD-1+LAG-3+ CD8+s, with low levels of proliferative markers (compared with the levels in other groups, Fig. 7a) and decreased cytokine levels compared with the combination regimen (Fig. 7b-e)."

Figure 7i does not show there to be more terminally exhausted T cells in either of the ICB treated groups compared to the control. They do show high levels compared to the combination regimen, so this only requires some rewording to clarify what appears to be the intended point.

In the Materials and Methods (section on IMOD implantation and treatments), it would be useful to include a description of how light was measured for delivery to tumor. The Results indicates that irradiation was performed to 50 mW/cm² for 20 sec. How was this irradiance measured? What light dose (J/cm²) was delivered to the tissue in the 20 sec?

Reviewer #3:

Remarks to the Author:

In the manuscript, the authors developed an implantable miniaturized device using electrode-embedded optical fibers (IMOD) with both local drug delivery (immune checkpoint blockade antibodies and photodynamic therapy agents) and tumor impedance measurement capabilities over the course of a few weeks. In vivo experiments, the device could delay tumor growth and elicited a sustained anti-tumor immune response in multiple tumor models. And the tumor impedance readings would synchronously respond to the change in tumor size. The IMOD the authors designed seemingly a very promising local delivery device, however, the tumor impedance

measurements of IMOD in different tumor types were conducted in subcutaneous tumor models, since the subcutaneous tumors can be easily surgically removed clinically, the authors should give a detailed comment and do supplementary experiment to further illustrate the clinical applicability and clinical significance of the device in deep tumors or intra-organ tumors. Another noteworthy issue is the injection times of ICB were not consistent with each other in each group in all the in vivo experiments (Figure 4-7 and related Figures in supporting information). For example, ICB in IMOD/PDT/ICB (daily) group were dosed daily while others were not, in my opinion, the comparison of each group should be meaningful when the drug dose and dosing schedules are consistent. Thus, the current results may not fully support the overall importance of this device. Other issues that need to be explained and discussed are detailed as follows.

1. As demonstrated in Figure 3, PDT could improve the intratumoral retention of proteins or ICB for approximately 1 week, why ICB antibodies in IMOD/PDT/ICB (daily) group were administered daily in all the in vivo experiments (Figure 4-7 and related Figures in supporting information)? Furthermore, more experiments' details should be illustrated in the methods section, such as drug dosage, time of administration and procedure of NIR illumination. For example, readers don't know the PDT is conducted only once at 4h after implantation or daily, if PDT is conducted only once, then why the authors coating 3 more PLA layers for the sustained release of verteporfin in Supplementary Figure 1b? Besides, I am confused about the IMOD/PDT (q3d x 2) group in Figure 6 and Figure 7 while absent in Figure 4, q3d x 2 means the dosing schedules of PDT or ICB?
2. In Figure 4a, the therapeutic effect of ICB (i.t q3d*4) was inferior than IMOD/ICB (q3d*4), then what's the advantages of the device compared with intratumoral injection? Please explain it and relative experiments should be conducted.
3. In Figure 3, authors claimed that PDT could improve the intratumoral retention of ICB antibodies for at least 6 days, does it only attributed to the enhanced tumor vessel permeation of therapeutics reported by other researchers? It would be amazing if PDT can enhance the retention of all the therapeutics to such a long time. So, what was the mechanism that Cy7-BSA stayed around the tumor site for such a long time, and what's the purpose of authors doing this experiment? Please discussed in detail and supplement necessary experiments.
4. As the authors mentioned that the device elicited a sustained anti-tumor immuneresponse in multiple tumor models, experiments about the abscopal therapeutic effect (e.g. lung metastasis) with immunotherapy should be conducted.
5. In Supplementary Figure 2a, verteporfin should be used as the fluorescent tracer instead of Rhodamine to directly and truly reflect the experimental results. In Supplementary Figure 2c, the isolated 4T1 tumor tissue was not enough to simulate the actual osmotic pressure of tumor, so the in situ tumor in living animal experiments should be conducted. In Supplementary Figure 4d, authors claimed that verteporfin stayed mainly within tumor tissue at 4h post-implantation, the fluorescent ruler and tumor location should be indicated.
6. The format of dates should be revised thoroughly. Such as scale bars of all images taken by microscope (Supplementary Figure 1c, 2b, 14, 15 and 16), the fluorescent ruler of the living imaging were missing.

Reviewer #1:

This manuscript developed the implantable optical fiber to release photosensitizer/ICBs for combined therapy. The authors show that this implantable optical fiber could reflect the therapeutic efficacy via tumor impedance. In this work, the combination between PDT and ICBs seems to exert good therapeutic efficacy as well as immune response. However, this efficacy may be just because of the very high frequency of ICBs injection, and some major issues need to be addressed.

Our mechanistic studies in Fig. 7 demonstrated that high frequency of ICB administration is necessary to prevent CD8⁺ TILs from terminal exhaustion (inactive and without anti-tumor immune function). In our discussion part, we also highlight that PDT could increase the progenitor exhausted CD8⁺ TILs functions compared with ICB antibodies. Daily intratumoral injection is practically improbable to perform, and therefore the use of our device is of preclinical relevance to locally administer immunotherapeutics for enhanced efficacy and avoid systemic toxicities. In the discussion section, we added:

“Additionally, our studies also demonstrate the necessity of continuous administration of ICB antibodies in order to prevent CD8⁺ TILs from terminal exhaustion (Fig. 7). The IMOD device is therefore advantageous in practice to repetitively provide local administration of toxic agents to tumors since daily intratumoral injection using syringe is not feasible.”

1. In figure 2, the authors didn't show how to calculate the normalized values, even in the experiment section. More importantly, the relationship between impedances and tumor sizes, but not their normalized values, should be more direct, convincing and meaningful. In addition, more discussions/explanations are necessary to illustrate the mechanism about the relationship between impedances and tumor sizes. Although the authors provided several references, it seems that the literature doesn't seem to support it.

Normalized value is calculated based on the starting value of the measurement (set as 1 for both size and impedance readings at 10 kHz). We note that in Method section. Our experimental results showed that we cannot use absolute values as each impedance readings after inserting into tumors, even for the similar sized tumors, vary significantly, and even cannot use average data to plot as deviations are large. We added Supplementary Figure 3d to show that point, where all tumor ~ 50-100 mm³ but tumor impedance values vary substantially.

This is also observed in the literature for ex vivo studies (our ref. 38 Cancer Res. 74, 6408-6418 (2014)). Nevertheless, for each individual tumor, once we set the initial readings as 1 (normalization), the trend between tumor sizes and impedance readings can be distinguished and plotted. Many literatures (ref. 35, 37 38, also ref. 6 in supplementary information section) do show the proportional relationship between the tumor size and impedance in ex vivo and in vitro studies, and no in vivo models have been established between them.

In our original manuscript Supplementary Table 1 and Supplementary Figure 18, we have discussed the mechanism in detail (with references) and also provide simulated data to support our impedance-tumor measurement model. To further elucidate the details, we added Circuit Model Discussion section after Supplementary Table 1. In brief, our circuit model consists of a resistance of the tumor interstitial fluid (TIF, R_{TIF}) in series with a constant phase element (CPE) representing double layer capacitor (Z_{DL}) and a parallel combination of the paracellular

resistance (R_{para}) and another CPE simulating the transcellular behavior of the cell (Z_{trans}). As IMOD was inserted in solid tumor when performing the electrochemical impedance spectroscopy, the effective thickness or the distance of the TIF that contributes to the R_{TIF} is relatively small. Moreover, since the ionic composition of TIF is very close to that of plasma, which has the electrical conductivity of 14.3 mS cm^{-1} at body temperature,^{10,11} R_{TIF} is negligible in the circuit which is further validated by the equivalent circuit fitting results shown in Supplementary Table 1. We also found that the paracellular resistance (R_{para}) and transcellular impedance ($1/A_{\text{trans}}$) were the critical components in the impedance composition. As shown in Supplementary Figure 18, we discovered that the normalized paracellular resistance and transcellular impedance increased linearly with the normalized tumor size, which correlated to our findings of the relationship between the normalized impedance value at a frequency of 10 kHz and the tumor size (Figure 2). This phenomenon can be further explained by the increase of the tumor cell number as tumor gets larger, leading to the proportional increase of the R_{para} and $1/A_{\text{trans}}$, which results in the proportional increase of the total impedance (detailed discussions and simulation results see Supplementary Figures 18-19 and Supplementary Table 1).

2. In figure 4, daily treatment of IMOD/PDT/ICB (ICBs: 5 mg/kg) is in a really high frequency, especially considering the claim from the authors that PDT prolonged the retention of ICB even up to 144 h in tumor (figure 3, line 81). The combination of free photosensitizer and ICB is a required control to confirm the advantage of IMOD/PDT/ICB.

As we mentioned above, the high frequency of administration is necessary to improve the efficacy. We added two groups, PDT (i.p. q3d \times 2) + ICB (i.p. q3d \times 4) and PDT (i.t. q3d \times 2) + ICB (i.t. q3d \times 4) in E0771 tumor models (see Supplementary Fig. 4e, due to the limited space in Figure 4a we cannot put too many lines in one plot). Neither prolonged the survival compared with IMOD/PDT \times 2 + ICB (i.p. q3d \times 4). We also note that even PDT prolonged the retention of BSA, the half-life of BSA is still 31 h (Figure 3c), which indicates that more ICB antibodies are still needed to keep CD8 T cells from exhaustion.

3. In figure 6, could the authors explain why anti-CTLA-4/anti-PD-1 in IMOD/PDT/ICB group could not block the CTLA-4/PD-1 in T cells, but increase the frequencies of CTLA-4+PD-1+ cells in CD8+ TILs?

Both PD-1 and CTLA-4 immune checkpoint molecules are expressed/upregulated few hours after CD8⁺ T cells are activated, and start to promote CD8⁺ T cells exhaustion. ICB antibodies bind to corresponding checkpoint molecules, preventing those checkpoint molecules from binding to CD28 and other ligands (PD-L1 etc.), which attenuate positive T cell co-stimulation signals (Cell. 170, 1120–1133 (2017)). The ICB antibodies cannot decrease or down-regulate the expression of CTLA-4 and PD-1. Instead, it is reported that the administration of ICB antibodies (or vaccines) does increase the frequency of double positive CD8⁺ T cells (see our ref. 8, PNAS 107, 4275–4280 (2010).), which indicates that more naïve CD8⁺ T cells are activated after treating with ICB antibodies. This also suggests the importance of continuously treating ICB antibodies in order to prevent CD8⁺ T cells from exhaustion. Above discussion was added after Supplementary Fig. 10.

4. Although the authors show the H&E images on day 7, a relative long term of biosafety of this device should be provided. Did the device induce the inflammation or cytokines in the interface between device and tissue?

As the fiber is directly inserted into tumor, it does not induce further inflammation (tumor tissue somehow is regarded as inflammation as it is fibrous, but lots of immunosuppressive macrophages and neutrophils and other immune cells inside). Notably, tumors were cured after 10-14 days receiving combination regimen in E0771 tumor-bearing mice and the device are removed from the mice. We added long-term (30 day) biosafety histology studies using 4T1-tumor bearing mice (see Supplementary Figure 17). We added “On day 30, no multinucleated foreign body giant cell was found around the place where IMOD fiber was inserted into the tumor (pointed by white arrow), presumably due to the immunosuppressive environment inside the tumor.”

5. In figure 5, tumor sizes would be better than their normalized values. Besides, did tumor impedance measurement respond to the ICB in terms of the mean value for five mice, but not just the value for individual mouse?

Same as question 1. For impedance measurement, absolute tumor size does not correlate with the absolute impedance value. For each tumor, different position has similar readings (Supplementary Figure 3a). However, the absolute values are significantly different for different tumors, even average numbers cannot be used for studies (Supplementary Figure 3d).

6. In figure 3, the fluorescence intensity in panel a at 0 h seems stronger than that in panel b. Can the authors show the absolute intensity, but not the normalized intensity in panel c?

We added the absolute intensity data in Supplementary Figure 2e. At 0h, panel a is similar to panel b but increase significantly at 1 h presumably due to the diffusion of dye in tumors. This is also observed in Supplementary Figure 2d where verteporfin was used.

7. In figure 1e, using rhodamine 6G to mimic hydrophobic drugs was not appropriate. The concentration of rhodamine 6G in water could be 1 mg/mL.

We thank the reviewer for pointing out this mistake. We used rhodamine B (which is documented in our original manuscript Methods section) instead of rhodamine 6G. Though the rhodamine B is still slightly soluble in water, its octanol/water partition coefficient ($\log K_{ow}$) of rhodamine B is 1.95, comparable to many hydrophobic molecules and drugs (coumarin: 1.39; doxorubicin (no salt form): 0.65; benzocaine: 1.89).

8. Please carefully check the manuscript, as there are many errors. Some descriptions in text are not consistent with the figures.

We thank the reviewer and we have carefully revised the manuscript.

Reviewer 2

The submitted manuscript describes IMOD (implantable miniature optical fiber device) as a technology to be used in the treatment solid tumors, via local delivery of both photosensitizer for photodynamic therapy (PDT) and antibodies for immune checkpoint blockade, as well as a tool to track tumor response (size) by impedance measurements. Data are presented on the immunological advantage of IMOD in stimulating anti-tumor immunity compared to either treatment as a standalone approach, and cured animals are demonstrated to reject a subsequent tumor rechallenge. A general comment relates to the explanation that is provided for why the combined approach is better than either modality as single treatment.

It is stated that PDT acts to cause tumor vessel permeation (section on Effect of PDT on intratumoral drug retention), which can increase tumor concentrations of drugs. This is true, but the present study utilized intratumoral administration of both the photosensitizer and ICB antibody. These drugs were not delivered systemically (e.g., i.v.), making the role of vascular effects less certain. Further clarification of how vascular permeability would be altered by IMOD or an alternate explanation for the expected interaction between ICB and PDT in the context of IMOD would be valuable here.

We thank the reviewer for pointing that out. The increased vascular permeability of tumor vessels not only works for i.v. vessels, but also improves the accumulation of drugs (especially proteins) via intratumoral injection, as the proteins with large size can be more easily repelled from the tumor bed due to the tumor solid and fluid stress. The enhanced vessel permeation thus allows the large molecules extravasation and stay in tumor bed. In addition, it is reported that the photodynamic therapy also caused reduction in tumor blood flow, which could also attribute to the prolonged retention of macromolecules. We added: “Additionally, previous studies showed that PDT could cause the reduction in tumor blood flow,⁵⁶⁻⁵⁸ which may also attribute to the prolonged retention of Cy7-BSA.”

Other specific comments include the following:

In supplemental Figure 2b, how are the images to be aligned with the photographs of Figure 2a? It is not clear what Figure 2b is intended to add in addition to that shown in 2a.

Supplementary Figure 2b showed the dye was well distributed inside the tumor instead of just locally concentrated or on the surface.

Supplemental Figures 1c, 1d and 14 do not appear to be called out in the text. Certain panels of Supplemental Figure 5 are also not called out and the relevance of the b cell studies is not mentioned in the text.

Due to the limit of space in the main text, we did not discuss every data in Supplementary information. We switched Supplementary Fig. 1b with 1c due to the description sequence. For Supplementary Figure 1b, we added “To avoid direct contact and prevent mechanical effect, transwell assays were set up to test the ability of tumor cells to uptake rhodamine B released from fibers, and we found that rhodamine B was effectively absorbed and retained by 4T1 murine mammary carcinoma cells (Supplementary Fig. 1b).”

For Supplementary Figure 1d, we added “4T1 breast cancer cell death was observed 20 hours after cells were incubated with a verteporfin-loaded fiber in a transwell and subjected to 20 s of

NIR light, while cells that were shielded for NIR light remained healthy, proving that verteporfin released from the fiber could be activated by NIR light (Supplementary Fig. 1d).”

We moved Supplementary Figure 14 as Supplementary Figure 10e (original Supplementary Figure 8). We added “...also see immunohistology images of intratumoral CD8⁺ cell in mice receiving the combination regimen in Supplementary Fig. 10e.”

For Supplementary Fig. 5 data, we mainly focus on the CD8 T memory cell’s impact on anti-tumor immunity. Role of B cells in tumor immunology is still controversial and under exploration. We added discussion section after Supplementary Figure 5 to discuss the data, due in part to the limited space in main text:

Though cured mice did not exhibit significant change in the fraction of CD8⁺ and CD4⁺ effector T cells in lymph nodes upon rechallenge (Supplementary Fig. 5a,b), a significant increase in both effector and central memory CD8⁺ T cells in the lymph nodes on day 90 was observed ($P = 0.001$ and 0.004 , respectively; $n = 6$ or 7 , Supplementary Fig. 5e, f). The increased fractions of T_{reg}s and MDSCs were likely due to immunosuppressive effects induced by injected tumor cells (Supplementary Fig. 5c, d). The increased fraction of double positive CTLA⁺PD-1⁺ in CD8⁺ T cells suggested that CD8⁺ T cells were activated to combat tumor cells (Supplementary Fig. 5g). The increased fractions of B cells may indicate its role in assisting CD8⁺ and CD4⁺ T cells against tumor.^{2,3}

What do the colors represent on Supplemental Figure 3a and 3b?

For Supplementary Figure 3a we added “Each color represents the independent study on each individual tumor volume.” For Supplementary Figure 3b we added “The color represents each independent impedance measurement.”

In Figure 2, it is stated that 6 mice were used for the measurements of panel a, 8 for panel b, 7 for panel c and 5 for panel d. This leads to confusion in the interpretation of the plots because tumor size does not seem to repeat itself as expected. Using panel a as an example, 84 data points from 6 mice suggests that there should be 14 data points plotted at each for 6 distant tumor volumes.

This is due to the tumor growth speed difference for different types of tumor. B16F10 tumor (panel d) grows much more rapidly compared to E0771 and 4T1 tumors (we start from 50-100 mm³ tumor and have to euthanize mice when tumor sizes reach 1000 mm³ based on IACUC protocols). Thus data points available to access vary significantly in different types of tumor.

What is the injected dose of verteporfin that 5ug is equivalent to?

We added “... which is equivalent to the dosage of 0.25 mg/kg verteporfin injected intravenously per mouse”. This is also the dose used in ref. 53.

In Figure 5/Supplemental Figure 6, please further clarify what the green highlighting is used to indicate. Why do the first set of measurements often go in opposite directions within this green box?

In both titles we have already mentioned that the green color highlight the correlation between tumor size shrinkage and impedance change. When tumor shrinks, impedance reading goes

lower. The tendency may not be exactly the same in every animal (e.g., Fig 5a third one) but correlations are obvious. The impedance may have 1-2 day delay responding to tumor size increase or decrease but trend reversal can be easily identified.

Figure 4b show that E0771 tumors had resolved by 10 days after the initiation of treatment. How was this dealt with in subsequent immunophenotyping when tumors were collected at this same timepoint for flow cytometry? What was available to assay if tumors had resolved?

We thank the reviewer for point out that. We collected tumors when their sizes were around 100 mm³ at day 8 after the start the treatment, and so we had to further isolate CD8⁺ TILs for better analysis. We added below notes in both Fig. 6 and main text: "...for mice receiving combination regimen, the collection of tumor started on day 8 while sizes were ~ 100 mm³."

Under the section on Combination therapy via IMOD impacts tumor immune infiltration, the text refers to Figure 6h as the ratio of MDSC to Teff cells instead of the ratio of Teff to MDSC, as shown in the figure.

We thank the reviewer pointing out the mistake and we changed to "CD8⁺ T_{eff} cells to MDSC"

Supplementary Figures 15 and 16 demonstrate that IMOD was not associated with toxicities, but do not show that it reduced toxicities as claimed in the first paragraph of the Discussion.

Comparison to a condition that caused toxicity was not performed, and the studies extended to only 7 days which is not a long period to justify that no toxicities were detected.

Similar to the reviewer 1's comment #4, we added Supplementary Figure 18 (using 4T1 tumor) to demonstrate that there is no toxicities over long period. Note in the E0771 tumor the device can be removed after the tumor is cured (10 -14 days). We note that in the literature of mice studies, the toxicity was documented (added in the main text) "... alanine aminotransferase (a liver damage marker) modestly increase upon repetitive intraperitoneal administration of ICB antibodies in mice.^{21,57}" However, on both day 7 and day 30, we did not observe any significant liver toxicity in mice. Note that no multinucleated foreign body giant cell was found around the place where IMOD fiber was inserted even on day 30.

In reference to the following statement in the Discussion: "However, treatment with ICB antibodies increased the number of CD8⁺ TILs (Supplementary Fig. 8a) but also resulted in high fractions of terminally exhausted subsets of CD8⁺ TILs (Fig. 7i) and dysfunctional PD-1⁺CD38⁺ and PD-1⁺LAG-3⁺ CD8⁺s, with low levels of proliferative markers (compared with the levels in other groups, Fig. 7a) and decreased cytokine levels compared with the combination regimen (Fig. 7b-e)." Figure 7i does not show there to be more terminally exhausted T cells in either of the ICB treated groups compared to the control. They do show high levels compared to the combination regimen, so this only requires some rewording to clarify what appears to be the intended point.

We thank the reviewer for pointing that out. We reword the sentence in order to clarify the comparison:

"Additionally, treatment with ICB antibodies increased the number of CD8⁺ TILs (Supplementary Fig. 10a), but resulted in low levels of proliferative markers (compared with the

levels in other groups, Fig. 7a), and high fractions of terminally exhausted subsets of CD8⁺ TILs, and dysfunctional PD-1⁺CD38⁺ and PD-1⁺LAG-3⁺ CD8⁺s (Fig.7k-l), with decreased cytokine levels compared to the combination regimen (Fig. 7b–e).”

In the Materials and Methods (section on IMOD implantation and treatments), it would be useful to include a description of how light was measured for delivery to tumor. The Results indicates that irradiation was performed to 50 mW/cm² for 20 sec. How was this irradiance measured? What light dose (J/cm²) was delivered to the tissue in the 20 sec?

We thank the reviewer for asking about the measurement method. In the method section, we added:

“To measure the light intensity of IMOD, a two-centimeter long IMOD was coupled to a silica fiber patchcord (Thorlabs) which was air-coupled to the NIR light source (Kessil H150-red LED light source of 34W, 600-700 nm). The light output was measured by a power meter (Thorlabs) with a photodetector (Thorlabs) attached, and the average power was 62.4 mW/cm² leading to a light dose of 1.25 J/cm² that was delivered to the tissue in 20 seconds.”

We also change the power intensity number to more accurate value in the main text.

Reviewer 3

In the manuscript, the authors developed an implantable miniaturized device using electrode-embedded optical fibers (IMOD) with both local drug delivery (immune checkpoint blockade antibodies and photodynamic therapy agents) and tumor impedance measurement capabilities over the course of a few weeks. In vivo experiments, the device could delay tumor growth and elicited a sustained anti-tumor immune response in multiple tumor models. And the tumor impedance readings would synchronously respond to the change in tumor size. The IMOD the authors designed seemingly a very promising local delivery device, however, the tumor impedance measurements of IMOD in different tumor types were conducted in subcutaneous tumor models, since the subcutaneous tumors can be easily surgically removed clinically, the authors should give a detailed comment and do supplementary experiment to further illustrate the clinical applicability and clinical significance of the device in deep tumors or intra-organ tumors.

We thank the reviewer’s good suggestions. To extend our studies from subcutaneous or transplantable tumors, we used female FVB/N-Tg(MMTV-PyVT)634Mul/J mice to perform efficacy studies (also known as MMTV-PyMT, mouse mammary tumor virus–polyomavirus middle T antigen), which spontaneously develop highly invasive ductal carcinomas in all 10 mammary fat pads with a high frequency of lung metastases (see Figure 4e and Supplementary Figure 7). The mice receiving combination regimen showed prolonged survival time compared to other groups, with lower total tumor burdens. These data indicate that our method could be applicable to genetically engineered mouse models of breast tumor that recapitulates several of the characteristics of virulent human breast cancer driven by strong oncogenes. Additionally, we also show that our treatment could elicit strong immunological memory against tumors, by treating cured mice with either subcutaneously or intravenously (tail-vein) injected tumor cells, and cured mice rejected tumor rechallenging in both cases (Supplementary Figures 5 and 6).

Therefore, our platform, though further optimization and development required, have substantial clinical relevance. By analogy to the efficacy in a genetically engineered mouse model with spontaneous arising tumors (Fig. 4e), such IMOD device could be applied *in situ* to vaccinate a human tumor driven by a strong oncogene before surgery in patients at high risk for the occurrence of metastatic disease, or in patients genetically prone to develop second primary cancers, such as those with inherited mutations (also see our cited references #82-84 about local immunotherapy's clinical relevance).

Another noteworthy issue is the injection times of ICB were not consistent with each other in each group in all the *in vivo* experiments (Figure 4-7 and related Figures in supporting information). For example, ICB in IMOD/PDT/ICB (daily) group were dosed daily while others were not, in my opinion, the comparison of each group should be meaningful when the drug dose and dosing schedules are consistent. Thus, the current results may not fully support the overall importance of this device.

We thank the reviewer for pointing that out. We noted that original IMOD/PDT/ICB (daily) is that mice were treated with PDT (q3d × 2) and ICB antibodies (daily) via IMOD, and clarified that throughout the manuscript. We also added more studies to make dose schedule consistent for comparison. We note some schedules are not feasible (e.g., i.t. daily of ICB) and we can only do i.t. q3d to avoid creating too many holes on mice tumors. We added i.p. daily of ICB in both E0771 and 4T1 tumors. See Figure 4 and Supplementary Figure 4e.

Other issues that need to be explained and discussed are detailed as follows.

1. As demonstrated in Figure 3, PDT could improve the intratumoral retention of proteins or ICB for approximately 1 week, why ICB antibodies in IMOD/PDT/ICB (daily) group were administrated daily in all the *in vivo* experiments (Figure 4-7 and related Figures in supporting information)?

Though BSA stayed over 1 week in tumor we note that even PDT prolonged the retention of BSA, the half-life of BSA is still 31 h (Figure 3c, also mentioned in main text). Another reason is what we have shown in Figure 7, after activation, CD8⁺ T cells quickly go exhaustion and lose functions, thus repetitive dose of ICB antibodies could keep CD8⁺ T cells at progenitor exhausted phenotype with anti-tumor polyfunctionality. As mentioned in the discussion, PDT also increased the progenitor exhausted phenotype fraction while only ICB antibodies eventually lead to terminally exhausted T cells.

Furthmore, more experiments' details should be illustrated in the methods section, such as drug dosage, time of administration and procedure of NIR illumination. For example, readers don't know the PDT is conducted only once at 4h after implantation or daily, if PDT is conducted only once, then why the authors coating 3 more PLA layers for the sustained release of verteporfin in Supplementary Figure 1b? Besides, I am confused about the IMOD/PDT (q3d x 2) group in Figure 6 and Figure 7 while absent in Figure 4, q3d x 2 means the dosing schedules of PDT or ICB?

We thank the reviewer for pointing that out. We noted that original IMOD/PDT/ICB (daily) is that mice were treated with PDT (q3d × 2) and ICB antibodies (daily) via IMOD, and clarified the schedule throughout the manuscript.

2. In Figure 4a, the therapeutic effect of ICB (i.t q3d*4) was inferior than IMOD/ICB (q3d*4), then what's the advantages of the device compared with intratumoral injection? Please explain it and relative experiments should be conducted.

These two groups did not have statistical significant difference in log-rank test ($P = 0.39$), which showing partial responses for the treatment as many other ICB antibodies studies. The device allows for daily injection without possibly creating multiple holes on the tumor site. In addition, daily intratumoral injection is not feasible. Overall, both treatments are less effective. We also tried daily i.p. injection of ICB antibodies (Supplementary Figure 4e; Figure 4c), and daily IMOD/ICB (Figure 4d), which showed less effective compared with combination regimen. All these studies suggested: (1) injection route is important, i.t. could be more effective than i.p (also see refs. 16, 17, 18, 60), (2) frequency is also important, as conventional i.t. cannot be practiced daily, and even with prolonged PDT, protein retention half-life was still only 31 hours. (3) As mentioned in first question, CD8⁺ T cell can be quickly exhaustion after activation and become terminally exhausted. Therefore only IMOD delivered ICB antibody assisted with PDT could improve the therapeutic efficacy, supported by our flow cytometry analysis (Figures 6-7).

3. In Figure 3, authors claimed that PDT could improve the intratumoral retention of ICB antibodies for at least 6 days, does it only attributed to the enhanced tumor vessel permeation of therapeutics reported by other researchers? It would be amazing if PDT can enhance the retention of all the therapeutics to such a long time. So, what was the mechanism that Cy7-BSA stayed around the tumor site for such a long time, and what's the purpose of authors doing this experiment? Please discussed in detail and supplement necessary experiments.

As mentioned above, though BSA stayed over 1 week in tumor we note that even PDT prolonged the retention of BSA, the half-life of BSA is still 31 h (Figure 3c, also mentioned in main text). Discussion in ref. 53 indicated macromolecules retention was affected more pronouncedly than small molecules after PDT treated tumors. We believe that both enhanced tumor vessel permeability and reduction of tumor blood flow attributed to such prolonged retention of Cy7-BSA, which was explained in the main text (we also discussed these in reviewer 2's first comments). As shown in our studies, without PDT, Cy7-BSA is quickly cleared out from the tumor (4 h half-life). As we mentioned in the introduction, one severe problem of intratumoral injection is the rapidly clearance rate of proteins from tumors, which could cause the toxicity and lower the efficacy. As we show in later flow studies, both PDT and ICB antibodies are required to prevent CD8 T cell terminally exhausted. All these experiments suggest that longer intratumoral retention is beneficial to improve efficacy. We also mentioned that because this was a proof-of-concept study, we did not thoroughly optimize the dosing or timing as current combination dose schedule have already shown efficacy in multiple aggressive tumor models.

4. As the authors mentioned that the device elicited a sustained anti-tumor immuneresponse in multiple tumor models, experiments about the abscopal therapeutic effect (e.g. lung metastasis) with immunotherapy should be conducted.

We thank the reviewer for pointing that out. We did tumor cell re-challenge in our original manuscript by subcutaneously injecting tumor cells (Supplementary Figure 5). To further demonstrate that our method elicits anti-tumor immunological memory, we intravenously injected E0771 tumor cells via tail-vein injection to mice cured by combination regimens. We found that cured mice also rejected the tumor growth in lung over 30 days after the injection (Supplementary Figure 7) while control mice all died because of lung metastasis. Flow cytometry analysis showed both effector and central CD8⁺ T memory cells fraction significantly increased in lymph nodes, similar to what we observed in mice rejecting subcutaneously injected tumor (Supplementary Figure 5). Note that the IACUC in our university did not allow us to subcutaneously implant two tumors to study the abscopal effect (treat one and monitor the other tumor change) as such method could increase burdens to mice.

5. In Supplementary Figure 2a, verteporfin should be used as the fluorescent tracer instead of Rhodamine to directly and truly reflect the experimental results. In Supplementary Figure 2c, the isolated 4T1 tumor tissue was not enough to simulate the actual osmotic pressure of tumor, so the in situ tumor in living animal experiments should be conducted. In Supplementary Figure 4d, authors claimed that verteporfin stayed mainly within tumor tissue at 4h post-implantation, the fluorescent ruler and tumor location should be indicated.

We agreed with the reviewer that Supplementary Fig. 2a and 2c were ex vivo studies and all have defects compared with in vivo studies. We did so in order to verify the ex vivo feasibility before going to in vivo studies as shown in Supplementary Fig. 2d and Fig. 3. We added the fluorescence ruler and tumor locations in Supplementary Fig. 2 accordingly. Note in Figure 2c 4T1 tumor, when incubated ex vivo, still have solid stress (which is pretty high, see PNAS 2012 109 (38) 15101-15108) that slows down molecule diffusion.

6. The format of dates should be revised thoroughly. Such as scale bars of all images taken by microscope (Supplementary Figure 1c, 2b, 14, 15 and 16), the fluorescent ruler of the living imaging were missing.

We added the fluorescence ruler and scale bars in Supplementary Fig. 2 accordingly.

Reviewers' Comments:

Reviewer #1:

Remarks to the Author:

The authors have provided more data to try to address the Reviewers' concerns. These data address some issues, but some specific issues still need to be addressed:

1. The injection times of ICB were not consistent with each other in each group in all the in vivo experiments, even after revision. For example, the ICB control groups are confusing among different tumor models in Figure 4, ICB (i.p. q3d × 4) and IMOD/ICB (daily) were used as controls in different figures. In addition, Figure S4e has shown that ICB administration (i.p. daily) is feasible. So IMOD/PDT×2 + ICB (i.p. daily) is a better control in all in vivo experiments, compared with "IMOD/PDT×2 + ICB (i.p. q3d × 4)", to confirm the advantages of IMOD/PDT×2/ICB (daily).
2. About the daily injection of ICBs (also mentioned by Reviewer 3), the authors claim this high-frequency injection was due to quick exhaustion of CD8+ T cells without daily injection as well as the half-life (31 h) of BSA. How do the authors calculate the half-life in Figure 3c and Figure S2e, setting the peak value, or fluorescence value at 0 h as 100%? The half-life in both conditions is over 48 h, but not 31 h, which means reduced injection frequency of ICB is possible.
3. About the long retention of BSA, in Figure 3c and Figure S2e, the difference of intensity value at 0 h between two groups is almost 2-fold, is there any reason for this? Does this difference contribute to the long retention, but not PDT? I am also confused by the great error bar difference between Figure 3c and Figure S2e. They are the same data and should show similar error bars at least.
4. In the text of Figure 6h, it should be "Ratio of CD8+ Teff cells to MDSC", but not "Ratio of MDSCs to CD4+CD25+Foxp3+ Treg cells".

Reviewer #2:

Remarks to the Author:

The revised manuscript has addressed numerous reviewer concerns. Newly added studies such as the FVB/N-Tg(MMTV-PyVT)634Mul/J mice model of spontaneous ductal carcinomas adds strength to the findings.

The Materials and Methods now indicates that treatment was initiated at a tumor size of 50 mm³ (Subcutaneous tumor injection) and treatment was discontinued when tumors shrunk below 75 – 100 mm³ (IMOD implantation and treatments). Please clarify as this does not seem possible.

The fluence rate and light dose of PDT that was delivered by IMOD have been clarified. What is the fluence rate and light dose that was delivered by externally delivered PDT in the comparative studies (information currently given is a lamp output of 34W for 20 sec)?

What distinguishes IMOD as a means of intratumoral delivery that makes it less effective than i.t. injection of ICB, comparing ICB and IMOD/ICB for q3d x4 (supplemental fig 4e)? Does increased frequency of ICB delivery by IMOD increase its effectiveness (for example, q2d dosing for E0771 was done for the impedance studies). Conversely, what makes IMOD/PDT more effective than external PDT in the treatments combined with i.p. ICB, given that the ICB is identical in these cases (supplemental fig 4e)?

Reviewer #3:

Remarks to the Author:

The authors conducted supplementary experiments to address part of our earlier concerns, the remaining concerns that haven't been fully addressed are listed as below:

1. We are happy that the authors added anti-tumor effect and immune mechanism evaluation of IMOD in FVB/N-Tg(MMTV-PyVT)634Mul/J mice. However, what we are most interested is whether IMOD has the capacity to read the tumor size of deep tumors or intra-organ tumors according to

the response to the tumor impedance. As there are many nanocarriers mediated PDT combined with immune checkpoints blockade have been reported, and they have achieved good therapeutic effects, the biggest highlight of this article is not in the evaluation of the therapeutic effect and mechanism of action, but the simultaneous monitoring of invisible tumors during treatment, and this should also be the reason why readers will be interested in the article in our opinion.

2. Though the authors have sufficiently attempted to address the concerns, but it is still not fully addressed. Supplementary Figure 4c showed that ICB (i.t. q3d*4) group had better effect than IMOD/ICB(q3d*4) group with significant difference observed, while no significant difference was observed between the two groups in Supplementary Figure 4a, please give a reasonable explanation.

3. Comment 5 has not been fully addressed, why not use verteporfin instead of rhodamine B as the fluorescence probe to truly and directly reflect the experimental results.

REVIEWER COMMENTS

Reviewer #1 (Remarks to the Author):

The authors have provided more data to try to address the Reviewers' concerns. These data address some issues, but some specific issues still need to be addressed:

1. The injection times of ICB were not consistent with each other in each group in all the in vivo experiments, even after revision. For example, the ICB control groups are confusing among different tumor models in Figure 4, ICB (i.p. q3d × 4) and IMOD/ICB (daily) were used as controls in different figures. In addition, Figure S4e has shown that ICB administration (i.p. daily) is feasible. So IMOD/PDT×2 + ICB (i.p. daily) is a better control in all in vivo experiments, compared with “IMOD/PDT×2 + ICB (i.p. q3d × 4)”, to confirm the advantages of IMOD/PDT×2/ICB (daily).

In Figure S4e, the group receiving ICB (i.p daily) did not show any significant delay on tumor growth. We followed the reviewer's suggestion of adding IMOD/PDT × 2 + ICB (i.p. daily) in updated Figure S4e, which confirms the advantages of IMOD/PDT×2/ICB (daily).

2. About the daily injection of ICBs (also mentioned by Reviewer 3), the authors claim this high-frequency injection was due to quick exhaustion of CD8+ T cells without daily injection as well as the half-life (31 h) of BSA. How do the authors calculate the half-life in Figure 3c and Figure S2e, setting the peak value, or fluorescence value at 0 h as 100%? The half-life in both conditions is over 48 h, but not 31 h, which means reduced injection frequency of ICB is possible.

We used exponential decay model to calculate the half time. The starting value is determined by the peak fluorescence signal, usually 4 h or 12 h time point. In the method section we added the calculation equations and methods:

“The analysis of the results was carried out using Origin software (Northampton, MA) by fitting the normalized fluorescence intensity (I) – time (t) curve into a exponential decay model according to equation (1).

$$I = a \cdot \exp(-t/T_1) + b \quad (1)$$

where a , b , T_1 are all constants fitted by Origin software. The starting point for the curve fitting was the peak value of the normalized fluorescence intensity, usually at 4 or 12 h. The obtained T_1 value was used to calculate the decay half-life τ based on equation (2).

$$\tau = \ln(2) \cdot T_1 \quad (2)$$

Note that the fluorescence intensity at $t = 0$ was normalized as 1.”

We would like to point out that the peak value is not 1 and the intensity at $t = 0$ was set as 1. The Figure R1 showed how we fitted the curve and obtained the results.

We also apologized for the mistakes in our previous calculation where we used the mean data to calculate the half-life, and not the half time of each curve, and that leads to increased value in the Cy7-BSA group. We corrected the results in the main text:

“The fluorescence signal for Cy7-BSA was observable in the tumor for approximately 1 week (median half-life, 33.8 hours; Fig. 3a, c). Conversely, Cy7-BSA administered through IMOD without prior PDT was quickly cleared from the tumors (median half-life, 8.9 hours; Fig. 3b, c).”

The updated results did not affect the conclusion in the main text.

Figure R1. The curve-fitting calculation of clearance half-life in Figure 3c. The data are shown for the group receiving IMOD/PDT/Cy7-BSA. The highlighted red line was T_1 value for the calculation of clearance half-life. Starting points for the fitting are 4 or 12 h based on the fluorescence peak intensities.

3. About the long retention of BSA, in Figure 3c and Figure S2e, the difference of intensity value at 0 h between two groups is almost 2-fold, is there any reason for this? Does this difference contribute to the long retention, but not PDT? I am also confused by the great error bar difference between Figure 3c and Figure S2e. They are the same data and should show similar error bars at least.

We did inject the same amount of Cy7-BSA in both groups. The fluorescence signals were actually low in both groups upon injection, due to self-quenching of the dye. We observed at initial 1-12 h post injection, the fluorescence intensity in tumor regions of both groups gradually increased over time, presumably due to diffusion of the dye throughout the tumor. The self-quenching phenomena were more pronounced in the group of free Cy7-BSA, likely because diameter of the needle (0.184 mm diameter for 28G needle) for i.t. injection is smaller than that of the IMOD fiber (~ 0.3-0.5 mm), and that the injected solution was more concentrated at the i.t. injection site. Note that verteporfin did not have any substantial absorbance at 800 nm wavelength that can affect the Cy7 fluorescence readings. We added such discussion after Figure S2.

We also note that Figure 3c is in log scale, and Figure S2e in linear scale according to the reviewer's request, which may cause visual difference between error bars. We attached the excel file for our imaging studies, including imaging raw data, normalized data, and the half-life time calculation.

4. In the text of Figure 6h, it should be "Ratio of CD8+ Teff cells to MDSC", but not "Ratio of MDSCs to CD4+CD25+Foxp3+ Treg cells".

We thank the reviewer for pointing out that mistake and we corrected Figure 6h accordingly.

Reviewer #2 (Remarks to the Author):

The revised manuscript has addressed numerous reviewer concerns. Newly added studies such as the FVB/N-Tg(MMTV-PyVT)634Mul/J mice model of spontaneous ductal carcinomas adds strength to the findings.

The Materials and Methods now indicates that treatment was initiated at a tumor size of 50 mm³ (Subcutaneous tumor injection) and treatment was discontinued when tumors shrunk below 75 – 100 mm³ (IMOD implantation and treatments). Please clarify as this does not seem possible.

As shown in Figure 4b, when we treated the tumor, the tumor size keeps increase above 100 mm³ and would not immediately shrink, even for daily treatment of IMOD/PDT/ICB. The description was actually reflected in Figure 4b where tumor would initially continue to grow when we started treatment, and we stopped treatment when tumor had shrunk from large tumor size of over 100 mm³ to small size of below 75-100 mm³

The fluence rate and light dose of PDT that was delivered by IMOD have been clarified. What is the fluence rate and light dose that was delivered by externally delivered PDT in the comparative studies (information currently given is a lamp output of 34W for 20 sec)?

We added in the method section:

“To measure the light intensity for externally applied photodynamic therapy, the same light source (away from the animal to avoid the heat’s impact) and measuring power meter were used. For a typical exposed tissue area for light irradiation with a diameter ~ 0.5 cm, the average power was 63.82 mW/cm², similar to that of IMOD.”

What distinguishes IMOD as a means of intratumoral delivery that makes it less effective than i.t. injection of ICB, comparing ICB and IMOD/ICB for q3d x4 (supplemental fig 4e)? Does increased frequency of ICB delivery by IMOD increase its effectiveness (for example, q2d dosing for E0771 was done for the impedance studies). Conversely, what makes IMOD/PDT more effective than external PDT in the treatments combined with i.p. ICB, given that the ICB is identical in these cases (supplemental fig 4e)?

As mentioned in our previous revision, the group of i.t. ICB (q3d × 4) and IMOD/ICB (q3d × 4) do not have statistically significant difference (also see our reply to reviewer 3’ second question). Visual difference may be due to the different group numbers, and the expanded mice studies confirmed no difference between two groups (Figure 4a-b). For aggressively growing tumors such as E0771, increased dose frequency did improve the efficacy. For IMOD/PDT×2/ICB (q2d), we also saw partial responses and data are updated in Figure S4e.

For IMOD/PDT with i.p. injected ICB, the tumor concentration of ICB antibodies will be significantly lower than those via locally administered ICB. The immunosuppressive tumor microenvironment requires constant reinvigoration of T cells to avoid terminally exhaustion. The i.p. injected ICB may just provide small portion of the ICB to tumor, and such amount may not be enough to reverse the T cell exhaustion course (also see recent studies comparing i.p and i.t distribution of ICB antibodies *Sci. Transl. Med.* 12, eaay3575 (2020)), as demonstrated by the group receiving daily i.p. ICB in Figure S4e. Our flow cytometry studies also showed that i.p. treatment with ICB antibodies increased the number of CD8⁺ TILs (Supplementary Fig. 10a), but resulted in low levels of proliferative markers (compared with the levels in other groups, Fig. 7a), and high fractions of terminally exhausted subsets of CD8⁺ TILs, presumably due

to the low concentration of ICB antibodies in tumors could not reverse the exhaustion of CD8⁺ TILs. We added the above discussion and references after Figure S4.

Reviewer #3 (Remarks to the Author):

The authors conducted supplementary experiments to address part of our earlier concerns, the remaining concerns that haven't been fully addressed are listed as below:

1、 We are happy that the authors added anti-tumor effect and immune mechanism evaluation of IMOD in FVB/N-Tg(MMTV-PyVT)634Mul/J mice. However, what we are most interested is whether IMOD has the capacity to read the tumor size of deep tumors or intra-organ tumors according to the response to the tumor impedance. As there are many nanocarriers mediated PDT combined with immune checkpoints blockade have been reported, and they have achieved good therapeutic effects, the biggest highlight of this article is not in the evaluation of the therapeutic effect and mechanism of action, but the simultaneous monitoring of invisible tumors during treatment, and this should also the reason why readers will interested in the article in our opinion.

We appreciate the reviewer's interest in IMOD reading of intra-organ or deep tumors, which is the next step of our project. In response, we would like to first point out that many reported nanoparticles with PDT and ICB, however, have not seen success in treating aggressively growing tumors (e.g., E0771 or MMTV-PyMT), as shown in Table R1 below. Many of them are tested on CT26 (Table R1 entries 1, 4, 6, 7, 11), which is actually easy to treat via ICB i.p q3d × 4 (see Supplementary Figure 4a). Some intratumorally injected particles for photo-ablation of tumors have been reported (entries 1-3, 7-9, 11), but we did not apply high-intensity light for photo-ablation (either PDT or photothermal therapy) to remove primary tumor. Some resected or photo-ablated tumors are administered with particles and ICBs to inhibit rechallenged tumor growth or tumor recurrence (entries 5, 6, 12). The truly impressive one is the reported study in *Sci. Immunol.* 2019, 4, eaau6584 (entry 10), wherein high intensity laser (1.2 W/cm² for 5 minutes) was still used but the treatment did effectively inhibit 4T1 tumor and slow B16F10 tumor growth by i.v. injected particles with light irradiation for photothermal therapy. Our data showed that changing the administration route (from systemic to local with high frequency or adjustable schedule) did significantly improve the therapeutic efficacy in the aggressively growing tumor models and genetically engineered mouse tumor models. We would also like to point out that many studies using particles and phototherapy superficially characterized the tumor immune responses, whereas we and many others have shown that not all intratumoral CD8⁺ cells function well in tumor, and these CD8⁺ cells most likely get exhausted and would not fight against tumor cells without constant activation in the immunosuppressive tumor microenvironments. The efficacy and mechanism of action using our newly developed local delivery method are indeed worth highlighting for the field. In the end, we are developing a platform for both therapeutic treatment and diagnostic measurement. We have to first show that our platform works well for both of its intended functions, and that the administered treatment works better than conventional systemic methods.

Table R1. Summary of studies using photodynamic therapy and ICB antibodies for tumor treatment.

Entry	Nanoparticles	ICB & other immunotherapeutics	Tumor model	Administration	Primary tumor	Abscopal effect	Rechallenge / recurrence	Reference
1	Fe-TBP MOF	anti-PD-L1	bilateral CT26	NPs: i.t. once + light, anti-PD-L1: q3d	>90% regression	>90% regression	100% against s.c. tumor rechallenge	J. Am. Chem. Soc. 2018, 140, 5670.
2	Cu-TBP nano MOFs	anti-PD-L1	bilateral B16F10	NPs i.t. once + light, anti-PD-L1: i.p. q3d	B16F10 Tumor growth inhibition indices of 98.3%, cure rate 33.3%	regressed distant tumors with a TGI of 94.9%	cured mice 100% against s.c. tumor rechallenge	Chem 2019, 5, 1892.
3	cationic W-TBP MOFs adsorbing CpG	CpG and anti-PD-L1	bilateral TUBO	NP: i.t. once + laser. anti-PD-L1: i.p. q3d.	>97 % tumor regression on day22	>97 % tumor regression on day22	N/A	Angewandte Chemie 2020, 132 1124-1128
4	core-shell NPs with oxaliplatin in the core and pyropheophorbide-lipid conjugate in the shell	anti-PD-L1	bilateral MC38 and CT26	MC38: i.p. q3d x 3 + light. CT26: i.p. every other day, for a total of two injections	significant growth delay in both tumor models	significant growth delay	N/A	Nat. Commun. 2016, 7, 12499.
5	Zn-pyrophosphate nanoparticles loaded with photosensitizer pyrolipid porphyrin-containing liposomal decorated with cetuximab and conjugated with IRDye800 and DOTA-Gd	anti-PD-L1	bilateral subcutaneous 4T1 and TUBO	i.v. NPs and i.p anti-PD-L1 q2d x 3 + light	100% elimination	92% reduction in 4T1 tumor size, significant inhibition of TUBO	N/A	J. Am. Chem. Soc. 2016, 138, 16686.
6	Upconversion particle loaded with Ce6 and R837	anti-PD-L1	CT26	NPs: i.v. once + light. anti-PD-L1: i.p. q3d 24hr after irradiation	complete tumor eradication	N/A	N/A	Nanoscale 2018, 10, 16738.
7	self-assembly of DSPE-PEG-maleimide and indocyanine green (ICG) onto UCNPs, followed by loading of the photosensitizer rose Bengal	anti-CTLA-4	bilateral CT26	NPs: i.t. once + laser. anti-CTLA-4: i.v. on day 9 and 13.	complete elimination	strong inhibition of distant tumor growth	complete inhibition	ACS Nano 2017, 11, 5, 4463-4474
8		anti-CTLA-4	bilateral 4T1	NPs: i.t. once on day8. anti-CTLA-4: i.p. on day 9,11,13	84% long-term survival	delayed growth	34% against s.c. tumor rechallenge	Advanced Science 2019, 6 1802157

9	polydopamine NPs coated with an upconversion layer of NaGdF ₄ :Yb/Er shell and modified with chlorin e6	anti-PD1	s.c. 4T1 and i.v. fLuc-4T1	NPs: i.t. once + laser. anti-PD1: i.v. on day 11,14,17.	77.8% long-term survival	N/A	metastasis inhibition	Advanced Materials 2019, 31 1905825.
10	MMP-2 sensitive αPDL1/ICG-based nanoparticle	anti-PD-L1	4T1, B16F10	i.v. once + light, start treatment when tumor reaches to 300mm ³	60% complete 4T1 tumor regression, prolonged survival of B16F10	N/A	progression of the secondary tumor was greatly suppressed	Sci. Immunol. 2019, 4, eaau6584.
11	Indocyanine green (ICG), R837 coencapsulated in PLGA NPs	anti-CTLA4 and R837	bilateral 4T1 and CT26	NPs i.t. once + laser, anti-CTLA4: i.v. at day 1,4,7.	complete thermal ablation	completely inhibited growth for the 4T1 model, and disappeared for the C26T model	nearly no metastasis of i.v. Fluc-4T-1 model and 90% of survival rate of s.c. fLuc-4T1	Nat. Commun. 2016, 7, 13193.
12	Hollow MnO ₂ nanoshells post modified with PEG and co-loaded with Ce6 and DOX	anti-PD-L1	bilateral 4T1 on day (-7)	NPs: i.v. once on day(-1) + laser on day0. anti-PD-L1: i.v. on day 1,3,5,7	delayed tumor growth	delayed tumor growth	N/A	Nat. Commun. 2017, 8, 902

Secondly, we are the first to prove that we could use such device for both local drug delivery (including PDT) and the reading of tumor impedance in response to treatment. Such theranostic device has never been reported and we have to first engineer and evaluate its performance *in vivo* in order to prove the value of our idea, before leaping into using it for deep tumor measurement. We have to point out that more engineering work (e.g., signal transduction, comfortability and compatibility) is needed for such prototype device to be used in deep tumors; wireless-based programmable signal measurement and communication are undergoing and require significant amount of engineering work for fitting into such miniature devices; and the mouse model is not enough for implanting such device deep inside an animal's body. Leapfrogging over our current bulk studies (this manuscript, and determining that it is neither important nor interesting) is both logistically and fundamentally unrealistic. We have to take steps as Rome wasn't built in a day.

Note that as this manuscript had taken a long time to revise, we added additional references to reflect the progress in this field.

2、 Though the authors have sufficiently attempted to address the concerns, but it is still not fully addressed. Supplementary Figure 4c showed that ICB (i.t. q3d*4) group had better effect than IMOD/ICB(q3d*4) group with significant difference observed, while no significant difference was observed between the two groups in Supplementary Figure 4a, please give a reasonable explanation.

In our previous revision, we noted that our statistical log-rank test showed there was no statistically significant difference between ICB (i.t. q3d × 4) and IMOD/ICB (q3d × 4). As shown in Figure R2a-c below, both groups had partial response to the ICB treatment. Fig. R2c showed the statistical test results. We understand that the reviewer may see the difference visually, and we realized that the group number differences between the two groups could lead to the visual difference. We then added more mice into

both groups so that they have the same group number ($n = 14$ for both in the updated data, including the previous ones). As shown in Fig. R2d-e, both groups still showed partial response to the treatment and there is no statistically significant difference between them. Note that in our previous revision, Figure 4a and Supplementary Figure 4e used the same data for the same treatment group. We also included the new data in Fig. R2d-e to update both Figure 4a and Supplementary Figure 4e.

Figure R2. Difference between the two groups in E0771 survival studies. (a) Survival curve in our previous revision for the groups of ICB (i.t. q3d \times 4) and IMOD/ICB (q3d \times 4). (b) Tumor growth curve in (a). (c) We used origin software to do the log-rank study and there is no significant difference between such two groups ($P = 0.69$). (d) Updated data for these two groups (which including the data in previous revision), and there is no statistical difference between the two groups (log-rank test, $n = 14$, $P = 0.91$). (e) Tumor growth curve in (d).

3、 Comment 5 has not been fully addressed, why not use verteporfin instead of rhodamine B as the fluorescence probe to truly and directly reflect the experimental results.

Verteporfin is relatively potent *in vitro*, which limits the monitoring time for *in vitro* or *ex vivo* studies. As shown in the updated Supplementary Fig. 2c, *ex vivo* cultured tumor disintegrated on day 4, presumably due to traces of light used in fluorescence imaging. We presented fluorescence images of day 1 to 4 and also the corresponding tissue slice of tumor sample obtained on day 1, as the tumor tissue had fell apart on day 4. Both data are updated in Supplementary Fig. 2.

Reviewers' Comments:

Reviewer #1:

Remarks to the Author:

The authors have addressed the previous reviews.

Reviewer #2:

Remarks to the Author:

The authors have addressed all of my outstanding questions. One further comment, which is more editorial in nature, pertains to the large number and often varied controls that are presented among the different panels in Figure 4 and Supplementary Figure 4. This makes it difficult to interpret these figures. One suggestion could be to create the figures with a reduced, but consistent set of controls that are included in every panel (noting here the request of another reviewer to provide a given control). Data from additional controls could be moved to a table. Another possibility could be to use bold typeface to identify the consistent set of conditions (including controls) that are carried across all of the experiments. Other approaches could also be used to simplify this presentation.

Reviewer #3:

Remarks to the Author:

I am satisfactory with the revision. The only suggestion is that the number of independent experiments performed should be included in each of the figure legends.

Reviewer #1 (Remarks to the Author):

The authors have addressed the previous reviews.

We thank the reviewer's positive comments.

Reviewer #2 (Remarks to the Author):

The authors have addressed all of my outstanding questions. One further comment, which is more editorial in nature, pertains to the large number and often varied controls that are presented among the different panels in Figure 4 and Supplementary Figure 4. This makes it difficult to interpret these figures. One suggestion could be to create the figures with a reduced, but consistent set of controls that are included in every panel (noting here the request of another reviewer to provide a given control). Data from additional controls could be moved to a table. Another possibility could be to use bold typeface to identify the consistent set of conditions (including controls) that are carried across all of the experiments. Other approaches could also be used to simplify this presentation.

We thank the reviewer's positive comments. We replaced the group names with the numbers in both figures for better presentation and comparison. All groups' names were added into the figure captions.

Reviewer #3 (Remarks to the Author):

I am satisfactory with the revision. The only suggestion is that the number of independent experiments performed should be included in each of the figure legends.

We thank the reviewer's positive comments. We added n numbers in figure legends.